# Transcriptomic and proteomic landscape of mitochondrial dysfunction reveals secondary coenzyme Q deficiency in mammals

Inge Kühl[1,2†]*, Maria Miranda[1†], Ilian Atanassov[3], Irina Kuznetsova[4,5], Yvonne Hinze[3], Arnaud Mourier[6], Aleksandra Filipovska[4,5], Nils-Göran Larsson[1,7]*

[1]Department of Mitochondrial Biology, Max Planck Institute for Biology of Ageing, Cologne, Germany; [2]Department of Cell Biology, Institute of Integrative Biology of the Cell (I2BC) UMR9198, CEA, CNRS, Univ. Paris-Sud, Université Paris-Saclay, Gif-sur-Yvette, France; [3]Proteomics Core Facility, Max Planck Institute for Biology of Ageing, Cologne, Germany; [4]Harry Perkins Institute of Medical Research, The University of Western Australia, Nedlands, Australia; [5]School of Molecular Sciences, The University of Western Australia, Crawley, Australia; [6]The Centre National de la Recherche Scientifique, Institut de Biochimie et Génétique Cellulaires, Université de Bordeaux, Bordeaux, France; [7]Department of Medical Biochemistry and Biophysics, Karolinska Institutet, Stockholm, Sweden

*For correspondence:
inge.kuhl@i2bc.paris-saclay.fr (IKü);
Larsson@age.mpg.de (N-GöL)

†These authors contributed equally to this work

Competing interests: The authors declare that no competing interests exist.

**Abstract** Dysfunction of the oxidative phosphorylation (OXPHOS) system is a major cause of human disease and the cellular consequences are highly complex. Here, we present comparative analyses of mitochondrial proteomes, cellular transcriptomes and targeted metabolomics of five knockout mouse strains deficient in essential factors required for mitochondrial DNA gene expression, leading to OXPHOS dysfunction. Moreover, we describe sequential protein changes during post-natal development and progressive OXPHOS dysfunction in time course analyses in control mice and a middle lifespan knockout, respectively. Very unexpectedly, we identify a new response pathway to OXPHOS dysfunction in which the intra-mitochondrial synthesis of coenzyme Q (ubiquinone, Q) and Q levels are profoundly decreased, pointing towards novel possibilities for therapy. Our extensive omics analyses provide a high-quality resource of altered gene expression patterns under severe OXPHOS deficiency comparing several mouse models, that will deepen our understanding, open avenues for research and provide an important reference for diagnosis and treatment.
DOI: https://doi.org/10.7554/eLife.30952.001

## Introduction

The energy from ingested food nutrients is harvested through a series of metabolic reactions and, in the final step, the mitochondrial oxidative phosphorylation (OXPHOS) system produces the cellular energy currency adenosine 5′-triphosphate (ATP). An adequate OXPHOS function is essential to support a variety of processes, such as biosynthesis of novel cellular components or cellular transport. Given the central role of mitochondria in cellular metabolism, mitochondrial dysfunction is a major contributor to human pathology and is also heavily involved in the aging process (*Larsson, 2010*). It is often assumed that the defects caused by OXPHOS dysfunction are directly explained by ATP deficiency. However, the network of metabolic reactions linked to OXPHOS is very complex and the molecular consequences of OXPHOS dysfunction are therefore hard to predict. Recent advances in

high-throughput technologies in proteomics, metabolomics, and sequencing have substantially increased our knowledge of mitochondrial function by refining the catalogue of mitochondrial proteins (*Calvo et al., 2016*), by broadening our understanding of the function of several orphan mitochondrial proteins (*Floyd et al., 2016*), and by identifying essential genes required for OXPHOS function (*Arroyo et al., 2016*). Furthermore, these studies have given novel insights into the pathophysiology of OXPHOS dysfunction by for example showing effects on intra-mitochondrial one-carbon (1C) metabolism necessary for purine and S-adenosine methionine synthesis (*Ducker and Rabinowitz, 2017*). However, our detailed understanding of the events that accompany OXPHOS dysfunction and contribute to the pathophysiology of mitochondrial diseases is still poorly understood and therefore the treatment options are very limited (*Pfeffer et al., 2012*).

The biogenesis of the OXPHOS system is under dual genetic control and requires the concerted expression of nuclear and mitochondrial DNA (mtDNA) encoded genes (*Gustafsson et al., 2016*). Mitochondria contain multiple copies of the compact double-stranded mtDNA, which encodes 2 ribosomal RNAs (mt-rRNAs), 22 transfer RNAs (mt-tRNAs), and 11 messenger RNAs (mt-mRNAs) producing 13 protein subunits of OXPHOS complexes I, III, IV and V. The biogenesis of the OXPHOS system is critically dependent on the mtDNA-encoded subunits as they typically have key catalytic roles or are core subunits for OXPHOS assembly (*Milenkovic et al., 2017*). Similar to the nuclear genome, expression of mammalian mtDNA requires several essential steps including genome maintenance, replication, transcription, RNA maturation and translation. All proteins involved in these processes are encoded in the nuclear genome, translated in the cytosol, and imported into the mitochondrial network. It is estimated that approximately one quarter of the ~1200 nucleus-encoded mitochondrial proteins are devoted to control mtDNA gene expression in mammals (*Gustafsson et al., 2016*). However, the regulation of mtDNA replication and gene expression still remains one of the most significant gaps in our understanding of mitochondrial function.

To better understand the cellular consequences of OXPHOS dysfunction in vivo we have performed a systematic comparative analysis of the mitochondrial proteome (mitoproteome) and the global transcriptome, as well as targeted metabolomics in five conditional knockout mouse strains with disruption of key genes that regulate mtDNA gene expression from mtDNA replication to translation in the heart. To abolish mtDNA replication, we disrupted the gene encoding the mitochondrial replicative helicase TWINKLE (*Twnk*; *Milenkovic et al., 2013*). TWINKLE forms a hexameric ring that unwinds mtDNA at the replication fork (*Fernández-Millán et al., 2015*; *Korhonen et al., 2004*). *TWINKLE* mutations in humans lead to multiple deletions of mtDNA, deficient respiratory chain function and neuromuscular symptoms. To study mtDNA maintenance, we disrupted the gene encoding mitochondrial transcription factor A (*Tfam*; *Larsson et al., 1998*). TFAM fully coats and packages mammalian mtDNA into mitochondrial nucleoids (*Kukat et al., 2015*, *2011*). It is also essential for mtDNA transcription initiation (*Shi et al., 2012*). To abolish mtDNA transcription, we disrupted the mitochondrial RNA polymerase (*Polrmt*) gene (*Kühl et al., 2014*; *Kühl et al., 2016*). To abolish mt-mRNA stability and maturation, we disrupted *Lrpprc* encoding the leucine-rich pentatricopeptide repeat containing protein that is required for posttranscriptional regulation (*Ruzzenente et al., 2012*). An amino-acid substitution in LRPPRC causes the French-Canadian type of Leigh syndrome, a severe neurodegenerative disorder characterized by complex IV deficiency (*Mootha et al., 2003*). Finally, to abolish mitochondrial translation we disrupted the mitochondrial transcription termination factor 4 (*Mterf4*) gene, which is necessary for a late assembly step of the mitochondrial ribosome (*Cámara et al., 2011*; *Metodiev et al., 2014*). These five genes are all essential as whole body knockouts are lethal at embryonic day 8.5. Tissue-specific deletion of these genes in the heart and skeletal muscle causes mitochondrial cardiomyopathy and a drastically shortened lifespan ranging from <6 weeks (*Polrmt* knockout) to 21 weeks (*Mterf4* knockout). Here, we combined these five models and their respective controls in a comparative study to systematically identify changes in levels of transcripts, proteins and metabolites due to disruption of mtDNA gene expression at different stages resulting in severe OXPHOS dysfunction. To study sequential protein changes during progressive mitochondrial dysfunction, we performed a temporal mitoproteomic analysis of the *Lrpprc* knockout mouse hearts at different ages. This allowed us to follow temporal events as the OXPHOS dysfunction progressed from mild to severe. In addition, we compared the transcriptomic and mitoproteomic changes of control mice to evaluate post-natal changes in the mitochondrial transcriptome and mitoproteome from juvenile mice (3-week-old mice) until adulthood (24-week-old mice).

Surprisingly, we identified that disruption of mtDNA expression has severe effects on the intra-mitochondrial coenzyme Q (ubiquinone, Q) synthesis. We found a global down-regulation of several key Q biosynthesis (COQ) enzymes accompanied by low levels of Q9, the most abundant form of Q in the mouse. These findings provide experimental rationale supporting Q supplementation as a general treatment for patients with OXPHOS dysfunction. Our label free proteomics approach is validated by the detection of very low levels of OXPHOS subunits in the studied mutants. Further, we reproduced previous reports that OXPHOS deficiency has drastic effects on the 1C metabolism and extend these findings by showing that up-regulation of the 1C metabolism is an early event present even before OXPHOS dysfunction is detected. Finally, our different high-quality extensive proteomic and transcriptomic datasets provide a rich source of information that can be mined for future studies of OXPHOS deficiency. They will be important resources for scientists and clinicians.

## Results

### Integrated RNA-sequencing and label-free mass spectrometry approach identify cellular and mitochondrial responses to OXPHOS dysfunction in mouse

We determined the mitoproteome and the cellular transcriptome in heart of five conditional knock-out mouse strains with impaired mtDNA gene expression at the level of replication (*Twnk*) (*Milenkovic et al., 2013*), mtDNA maintenance (*Tfam*) (*Larsson et al., 1998*), transcription (*Polrmt*) (*Kühl et al., 2014*, *Kühl et al., 2016*), RNA stability (*Lrpprc*) (*Ruzzenente et al., 2012*) and translation (*Mterf4*) (*Cámara et al., 2011*). All of these knockouts develop progressive OXPHOS deficiency and cardiomyopathy that leads to a much shorter lifespan ranging from <6 to 21 weeks (*Figure 1A*; *Table 1*). Two of the mouse models showed an increase in citrate synthase activity, which is a commonly used marker of mitochondrial biogenesis (*Figure 1—figure supplement 1*).

To determine the mitoproteome we isolated ultrapure mitochondria from end-stage hearts of all knockouts and used label-free mass spectrometry to identify differentially expressed proteins between knockouts and age-matched controls (*Figure 1B*). In parallel, we did RNA sequencing (RNA-Seq) on total RNA from end-stage hearts of the different knockouts. The reproducibility of the biological replicates was tested by Pearson correlation and visualized in a correlation heatmap for the transcriptome and proteome profiles (*Figure 1—figure supplement 2A*). Overall, the transcriptomic and mitoproteomic data were highly reproducible with a correlation coefficient of >85% across all samples and >95% between biological replicates. In total, 1118 transcripts encoding mitochondrial proteins were quantified from RNA-Seq. The proportion of mitochondrial proteins over the total protein mass in all the samples was ~99% and from the ~750 quantified proteins ~650 were mitochondrial based on MitoCarta 2.0 (*Calvo et al., 2016*; *Figure 1—figure supplement 2B*). Despite the comprehensive quantification and enrichment of mitochondrial proteins, we identified a significant bias against the detection of low abundant proteins in the mitoproteomes, which is a common limitation of label-free mass spectrometry techniques. We did not find differences in the detection of mitochondrial proteins based on their hydrophobicity or isoelectric point (*Figure 1—figure supplement 3*). Importantly, the distribution of fold changes compared to the label-free quantification (LFQ) intensity of controls or knockouts revealed proteins where the fold change estimations are largely influenced by imputed values in one of the genotypes as they are only detected in one genotype but not the other (eg. Delta-1-pyrroline-5-carboxylate synthase (*Aldh18a1*) and pyrroline-5-carboxylate reductases 2 (*Pycr2*); *Figure 1—figure supplement 4*). The effects of imputed values for individual proteins can be assessed in the *Supplementary file 8*. Differentially regulated mitochondrial genes and proteins were manually classified into 19 functional categories.

These datasets were complemented with a temporal mitoproteomic analysis of conditional *Lrpprc* heart knockout mice at 2, 3, 5, 7 and 10 weeks of age. In addition, we generated a third dataset comparing the RNA-Seq and proteomic data from all control hearts to evaluate post-natal changes in the transcriptome and mitoproteome of juvenile mice (3 weeks) at different time points until adulthood (24 weeks) (*Figure 1B*).

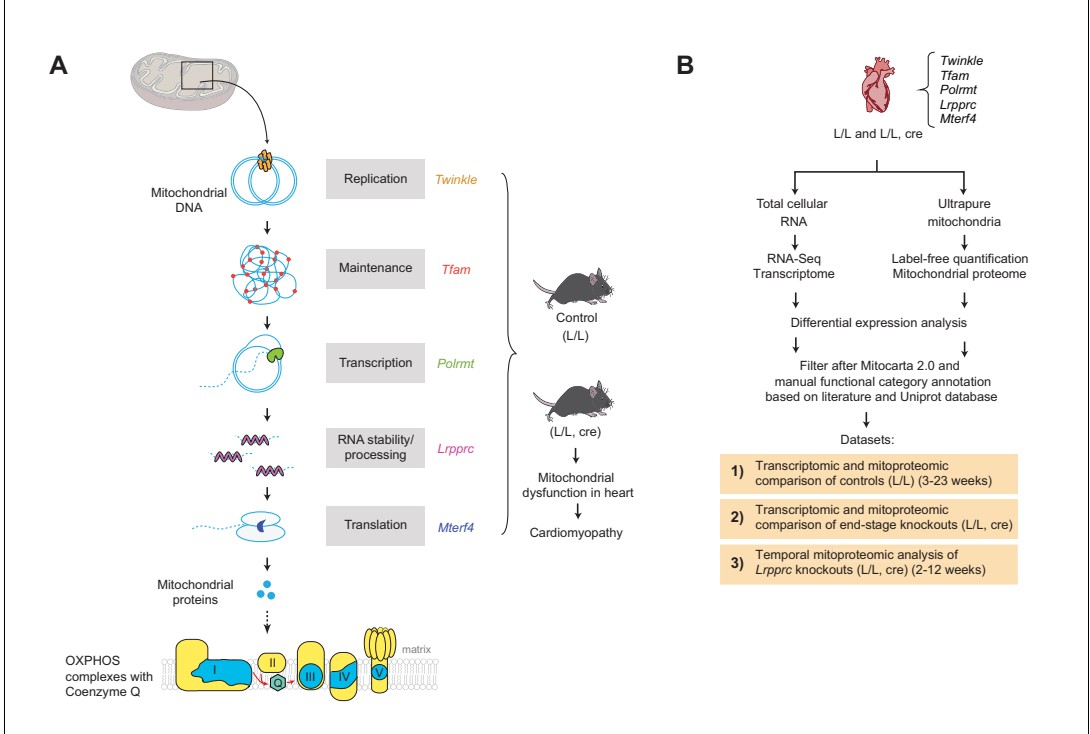

**Figure 1.** Label-free quantification and comparison of mitoproteomes of mouse hearts with mitochondrial dysfunction. (**A**) Schematic representation of the tissue-specific knockout strains (L/L, cre) used, with impaired mtDNA gene expression leading to mitochondrial dysfunction and corresponding controls (L/L). Nuclear-encoded mitochondrial proteins are illustrated in yellow and mtDNA-encoded mitochondrial proteins in blue. (**B**) Experimental workflow of the analysis of the transcriptomes and mitoproteomes from mouse heart generating three distinct datasets (1-3). Files are provided in *Supplementary file 1–5*, *8* and *9*.

DOI: https://doi.org/10.7554/eLife.30952.002

The following source data and figure supplements are available for figure 1:

**Figure supplement 1.** Activity of citrate synthase in heart from different L/L, cre and L/L mice.

DOI: https://doi.org/10.7554/eLife.30952.003

**Figure supplement 1—source data 1.** Determination of citrate synthase activity.

DOI: https://doi.org/10.7554/eLife.30952.007

**Figure supplement 2.** High reproducibility of the transcriptomic and mitoproteomic data.

DOI: https://doi.org/10.7554/eLife.30952.004

**Figure supplement 3.** Analysis of systematic bias in the detection of mitochondrial proteins.

DOI: https://doi.org/10.7554/eLife.30952.005

**Figure supplement 4.** Distribution of label-free quantification (LFQ) intensities and fold changes of quantified proteins.

DOI: https://doi.org/10.7554/eLife.30952.006

## Post-natal development of mouse heart is regulated at the protein level

Wild-type mouse tissues show a rapid post-natal increase in mtDNA from 1 to 7 weeks of age (*Chen et al., 2010*; *Figure 2—figure supplement 1A*) and factors required to maintain and express mtDNA follow this trend (*Figure 2—figure supplement 1B*). From a total of 756 quantified mitochondrial proteins, 51.2% changed significantly with age in control mouse heart (*Figure 2A*) consistent with post-natal development. Interestingly, these changes were not reflected at the transcript level where only 14% of the mitochondrial transcripts changed with age in control hearts (*Figure 2B*).

We performed clustering analysis to study how protein levels change with post-natal age in controls and found that 91.3% of the significantly changed proteins were distributed into four main clusters with different patterns (*Figure 2C*; *Figure 2—figure supplement 1C*). Clusters 1 and 4 contained the majority of mitochondrial proteins and interestingly they showed opposite patterns. Cluster one was enriched in proteins involved in pyruvate and lipid metabolism that increase until

**Table 1.** Summary of Major Characteristics of the Five Different Tissue-Specific Knockout Mouse Strains.

Arrows: increase or decrease; tilde: stable. Conditional knockouts: $Twnk^{loxP/loxP}$, $+/Ckmm-cre$ (**Milenkovic et al., 2013**), $Tfam^{loxP/loxP}$, $+/Ckmm-cre$ (**Larsson et al., 1998**), $Polrmt^{loxP/loxP}$, $+/Ckmm-cre$ (**Kühl et al., 2014**; **Kühl et al., 2016**), $Lrpprc^{loxP/loxP}$, $+/Ckmm-cre$ (**Ruzzenente et al., 2012**), $Mterf4^{loxP/loxP}$, $+/Ckmm-cre$ (**Cámara et al., 2011**).

| Conditional knockout (L/L, cre) | Twnk | Tfam | Polrmt | Lrpprc | Mterf4 |
|---|---|---|---|---|---|
| Gene product | Mitochondrial DNA helicase TWINKLE | Mitochondrial transcription factor A | Mitochondrial RNA polymerase | Leucine-rich pentatricopeptide repeat containing protein | Mitochondrial transcription termination factor 4 |
| Lifespan (weeks) | < 19 | < 10 | < 6 | < 16 | < 21 |
| Mitochondrial cardiomyopathy | + | + | + | + | + |
| Mitochondrial DNA | ↓ | ↓ | ↓ | ~ | ↑ |
| Mitochondrial DNA transcripts | ↓ | ↓ | ↓ | ↓* | ↑ |
| OXPHOS | ↓ | ↓ | ↓ | ↓ | ↓ |

* except 12S mt-rRNA, 16S mt-rRNA, mt-Nd6 and most mt-tRNAs

DOI: https://doi.org/10.7554/eLife.30952.008

reproductive maturity, whereas cluster 4 contained 90% of the mitochondrial ribosomal proteins that decrease until 8 weeks of age. 2D annotation enrichment analysis (**Cox and Mann, 2012**) of the mitochondrial transcripts and proteins showed that the drastic changes on the mitoribosomal proteins likely are explained by reduced protein stability (**Figure 2D,E**).

Notably, ~50% of the significantly changed OXPHOS proteins are decreased between 8 and 12 weeks of age (cluster 2). Taken together there seems to be a switch at ~8 weeks of age in controls when there is a boost in levels of key mitochondrial metabolism proteins whereas proteins involved in mtDNA expression and maintenance drop.

## *Atf4* and *Myc* transcription factors are increased in mouse hearts with severe mitochondrial dysfunction

Canonical pathway analysis of the differentially regulated genes in all knockouts identified by RNA-Seq, showed enrichment of several signaling and metabolic pathways (**Figure 3A**). Interestingly, the most enriched pathway was signaling via eukaryotic translation initiation factor 2 (eIF2), mediated by the cyclic AMP-dependent transcription factor ATF-4 (*Atf4*), myc proto-oncogene protein (*Myc*), C/EBP-homologous protein (*Chop/Ddit-3*) and cyclic AMP-dependent transcription factor ATF-5 (*Atf5*). Furthermore, *cis*-regulatory sequence analyses using iRegulon (**Janky et al., 2014**) predicted both, *Atf4* and *Myc*, among the top regulators based on our datasets (**Supplementary file 6**). *Atf4* and *Myc* have previously been associated with mitochondrial biogenesis (**Morrish and Hockenbery, 2014**) and we verified their expression levels via RT-qPCR. The *Myc* and *Atf4* transcription factors showed a very strong increase in all five knockouts in contrast to more general transcription factors involved in mitochondrial biogenesis, such as nuclear respiratory factor 1 (*Nrf1*), peroxisome proliferator-activated receptor gamma coactivator 1-alpha (*Pgc1a*) and GA-binding protein alpha chain (*Gabpa*) (**Figure 3B**). In line with these findings, several target genes of ATF4 and MYC are regulated at the transcript level (**Figure 3—figure supplement 1**) including some of the most upregulated genes encoding mitochondrial proteins such as mitochondrial bi-functional methylenetetrahydrofolate dehydrogenase/cyclohydrolase (*Mthfd2*), pyrroline-5-carboxylate reductase 1 - mitochondrial (*Pycr1*), tubuline beta 3 chain (*Tubb3*) and NADH-cytochrome b5 reductase 1 (*Cyb5r1*; **Figure 3C,D**; **Figure 4—figure supplement 1**).

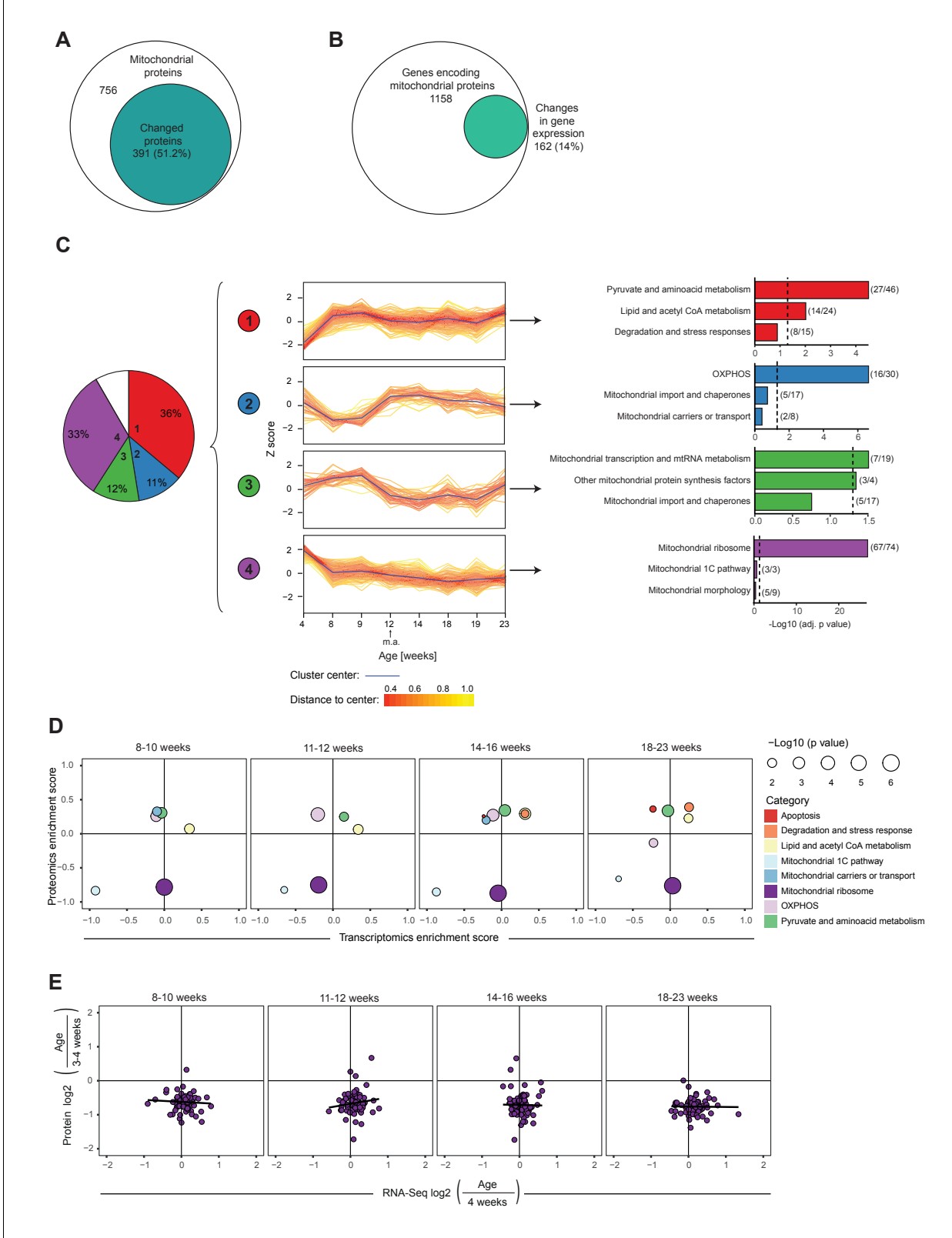

**Figure 2.** Mitochondrial transcriptome and proteome during post-natal development of mouse heart. (**A–B**) Venn diagram of significantly changed (**A**) mitochondrial proteins and (**B**) genes encoding for transcripts of mitochondrial proteins of L/L mice (3–20 weeks). (**C**) Hierarchical clustering analysis of mitoproteomes of L/L mice (3–20 weeks). Left to right: pie chart illustrating percentages of significantly changed mitochondrial proteins in each cluster (1-4), in white: not classified (8%); protein changes over ages for each cluster, fold change relative to 3 week old mice were scaled and presented as

*Figure 2 continued on next page*

*Figure 2 continued*

Z-score; top three enriched categories of each cluster. Dotted line: Benjamini-Hochberg adjusted p=0.05. Parentheses indicate the number of proteins changed in that category per total number of proteins classified in that category. m.a. = mature adulthood. (D) Mitochondrial transcriptomic and proteomic 2D enrichment analysis showing enriched functional categories of L/L mice at different ages compared to weeks 3–4 based on the fold change. (E) Correlation plots of the L/L, cre versus L/L fold changes between mitoribosomal transcripts and proteins in L/L mice at different ages. Black line indicates the trend.

DOI: https://doi.org/10.7554/eLife.30952.009

The following source data and figure supplements are available for figure 2:

**Figure supplement 1.** Rapid post-natal increase of mtDNA levels and factors required to maintain and express mtDNA in young wild-type mouse heart.

DOI: https://doi.org/10.7554/eLife.30952.010

**Figure supplement 1—source data 1.** qPCR determination of mtDNA levels in wild type mice.

DOI: https://doi.org/10.7554/eLife.30952.011

**Figure supplement 1—source data 2.** Densitometry analyses of western blots on total proteins.

DOI: https://doi.org/10.7554/eLife.30952.012

## Mitochondrial dysfunction causes strong post-transcriptional regulation of the mitoproteome

To identify common hallmarks of mitochondrial dysfunction in mouse heart we compared the transcriptomes and mitoproteomes of the five knockouts with severe mitochondrial dysfunction. Among the genes encoding mitochondrial proteins, we identified 420 genes (38%) that were differentially expressed in all knockouts, and most of these transcripts were decreased (*Figure 4A*; *Figure 4—figure supplement 1A*). In contrast, the majority (~65%) of the quantified mitochondrial proteins showed significantly changed levels, and most of these were increased in abundance (*Figure 4A*; *Figure 4—figure supplement 1B*). The transcriptome and mitoproteome are thus strongly affected in response to severe mitochondrial dysfunction, but the changes in the mitoproteome do not reflect the changes at the transcript level. Therefore, we conclude that the mitoproteome in mouse heart is mainly regulated at the protein level in response to severe mitochondrial dysfunction.

To further study the differences between the mitochondrial transcriptome and proteome during severe mitochondrial dysfunction, genes encoding mitochondrial proteins were included in downstream analyses if they changed significantly in at least one knockout strain. Overall, we found 310 genes whose expression changed exclusively at the transcript level, 186 genes at the protein level, and 470 genes at both the transcript and protein levels (*Figure 4B*). Interestingly, the correlation in differential expression of transcripts and proteins from nuclear genes encoding mitochondrial proteins was 32–56%, with higher correlation values in the *Lrpprc* knockout strain (*Figure 4C*; *Figure 4—figure supplement 1C*).

Next, we performed 2D annotation enrichment analysis of the degree of regulation of mitochondrial transcripts and proteins using our functional categories in all knockouts. There was a concordant up-regulation of apoptosis, degradation and stress response, mitochondrial import and chaperones, and the mitochondrial 1C pathway, whereof the latter had the highest enrichment scores in both dimensions (*Figure 4D*). Other mitochondrial protein synthesis factors, showed an anti-correlative behavior with increased proteomic enrichment scores. OXPHOS was more decreased in the proteomic compared to the transcriptomic dimension in all knockout strains (*Figure 4D,E*).

Transcript levels from nuclear genes encoding subunits of complexes I-V were slightly decreased in all knockouts, whereas there was a severe decrease in levels of proteins of complexes I, III, IV and V encoded by both, mtDNA and nuclear DNA (*Figure 5A*). The nucleus-encoded subunits of complex V, which can form a stable sub-assembled $F_1$ complex in response to severe OXPHOS dysfunction (*Mourier et al., 2014*), and the subunits of the exclusively nucleus-encoded complex II were slightly increased or unaffected in all knockouts. Interestingly, proteins involved in OXPHOS assembly were mainly increased with particularly high levels of the complex IV assembly proteins COX15 and SCO2 (*Figure 5B*), likely a compensatory response to the dramatic decrease in OXPHOS polypeptides. In general, RNA levels of genes encoding for proteins involved in OXPHOS assembly were less affected except for the complex IV assembly factor 6 homolog (*Coa6*) that showed a marked decrease in the *Tfam* and *Polrmt* knockouts. Taken together, these findings show that the drastic decrease in steady-state levels of nuclear-encoded OXPHOS subunits is a consequence of reduced

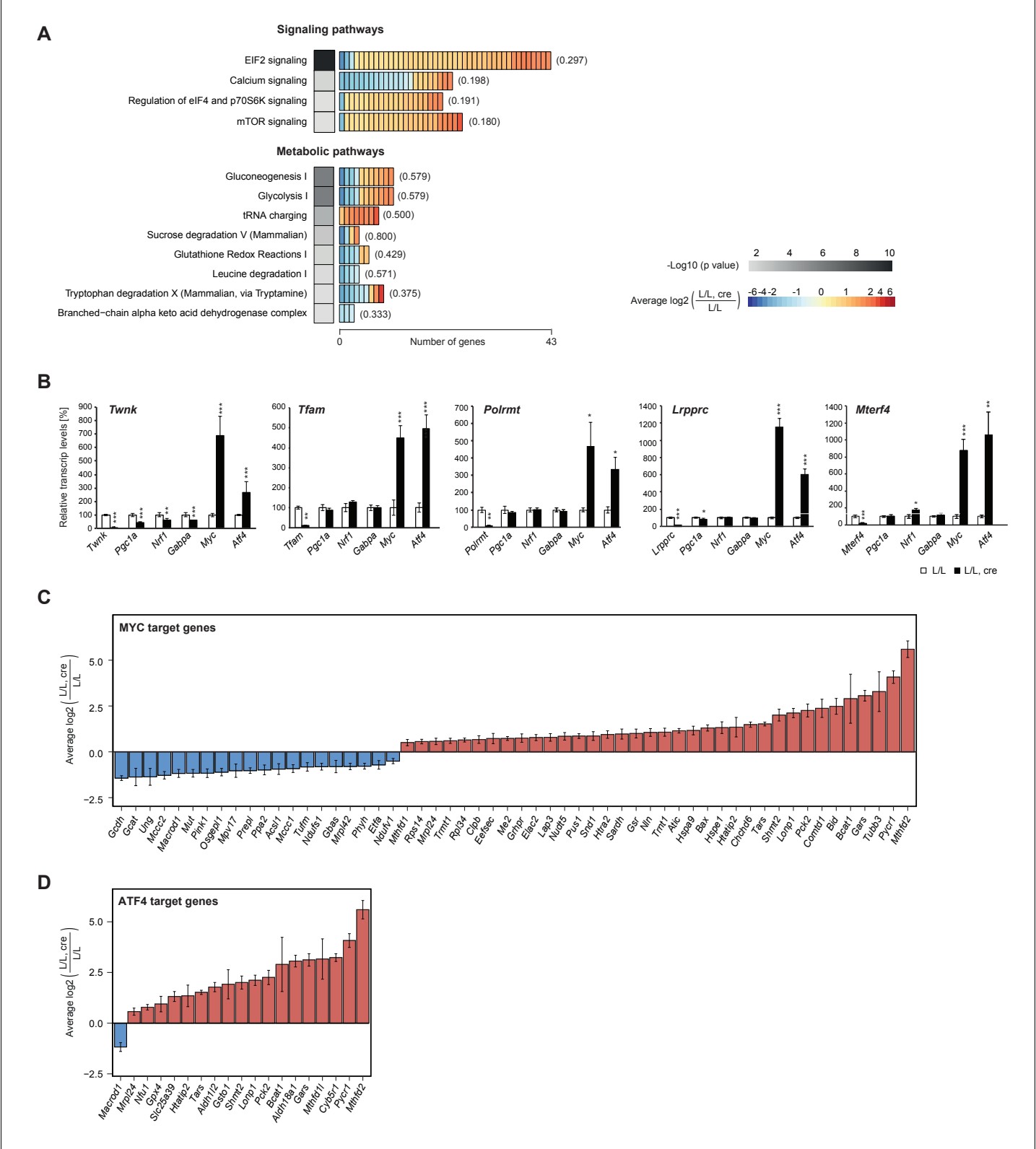

**Figure 3.** Enrichment of signalling and metabolic pathways in mouse hearts with severe mitochondrial dysfunction. (**A**) Canonical pathway analysis of significantly changed genes in all knockouts with representation of the 12 most significant pathways. Grayscale heatmap: p value of each pathway based on Fisher's exact test. Rectangles in horizontal heatmaps: average expression level in the five knockouts of each gene detected per pathway; Parenthesis: fraction of genes detected per pathway. (**B**) Transcript levels of genes encoding transcription factors involved in mitochondrial biogenesis

*Figure 3 continued on next page*

*Figure 3 continued*

in L/L, cre and L/L hearts. Normalization: B2M (beta-2-microglobulin). *Twnk, Tfam, Polrmt, Lrpprc,* and *Mterf4* were used as controls for the corresponding knockout strains. (**C–D**) Expression levels of differentially expressed MYC and ATF4 target genes encoding mitochondrial proteins. Graphs average expression level in the five knockouts of each gene ± SD.

DOI: https://doi.org/10.7554/eLife.30952.013

The following source data and figure supplement are available for figure 3:

**Source data 1.** qRT-PCR of genes encoding mitochondrial biogenesis factors in the five knockout mouse strains.

DOI: https://doi.org/10.7554/eLife.30952.015

**Figure supplement 1.** Several targets of MYC and ATF4 transcription factors are differentially regulated upon mitochondrial dysfunction.

DOI: https://doi.org/10.7554/eLife.30952.014

protein levels and not due to decreased mRNA expression. Furthermore, the comparison between proteomic and transcriptomic analyses that we present here argue that the regulation of intra-mitochondrial protein levels plays a major role in regulation of biogenesis of the OXPHOS system.

The abundance of mitochondrial ribosomal (mitoribosomal) proteins was dependent on the level at which mtDNA gene expression was disrupted. Knockouts with reduced steady-state levels of mt-rRNAs, that is *Twnk, Tfam* and *Polrmt*, showed massively reduced levels of mitoribosomal proteins, whereas knockouts with increased mt-rRNA levels, that is *Lrpprc* and *Mterf4*, had increased levels of mitoribosomal proteins (*Figure 5C*). These findings are consistent with the model that mt-rRNAs stabilize the nucleus-encoded mitoribosomal proteins. RNA-Seq showed that the levels of mRNAs from nuclear genes encoding mitoribosomal proteins were mildly affected and therefore did not explain the mitoribosomal protein levels (*Figure 4E*). Since the mitoribosomal proteins represent a significant proportion of the mitochondrial proteome, we repeated the correlation analysis in all knockouts after exclusion of these proteins (*Figure 4C*). It then became apparent that the lower correlation between the transcriptome and proteome in the *Twnk, Tfam* and *Polrmt* knockouts was explained by the abundant mitoribosomal proteins (*Figure 4—figure supplement 2*).

The pathways for degradation, stress response, mitochondrial morphology, apoptosis, protein import, and iron sulphur cluster and heme biogenesis (*Figure 5D,E*; *Figure 5—figure supplement 1*) showed an increase at the protein level. A particular strong increase (up to 8-fold) was detected in the expression of proteins involved in mitochondrial-mediated apoptosis (e.g. the apoptosis regulator, BAX, the mitochondrial DIABLO homolog, DIABLO, and the serine protease, HTRA2) and degradation and stress responses (e.g. AFG3-like protein 1, AFG3L1 and Lon protease, LONP1). The increased levels of AFG3L1, LONP and ATP-dependent Clp protease ATP-binding subunit ClpX-like (CLPX) correlate well with the decrease in protein subunit levels of OXPHOS complexes (*Figure 5A*). Interestingly, there was a concomitant strong increase in levels of transcripts for some of these factors, such as the *Lonp1, Bax* and BH3-interacting domain death agonist (*Bid*; *Figure 5D,E*), suggesting that the increased expression is due to a nuclear transcriptional response. Since several of the proteins involved in degradation, stress response or apoptosis are encoded by target genes of ATF4 and MYC (*Figure 3C,D*), we analysed the transcriptional regulation by the two transcription factors. We coloured the mitochondrial target genes of either ATF4 or MYC in the scatterplots of the RNA-Seq versus the proteomics data of all the knockouts of the different categories and we detected clear changes at the transcript level for some of the categories such as apoptosis, degradation and stress response (*Figure 4—figure supplement 3*).

It has been reported in *Caenorhabditis elegans* that mitochondrial stress triggers an unfolded protein response (UPRmt), a signaling cascade that is mediated by the transcription factor ATFS-1 (*Qureshi et al., 2017*). Although in mammals this response remains controversial (*Seiferling et al., 2016*), UPRmt is suggested to be mediated by CHOP and the ATF family of transcription factors, specifically ATF5 (*Qureshi et al., 2017*), which promote the expression of stress response genes such as mitochondria-specific chaperones 60 kDa heat shock protein (*Hsp60/Hspd1*), 10 kDa heat shock protein (*Hspe1*) and mitochondrial stress-70 protein (*mtHsp70/Hspa9*). We found an increase in protein levels of HSPD1, HSPE1, and HSPA9 in all five knockouts (*Figure 5D*; *Figure 5—figure supplement 1B*; box plots in *Supplementary file 8*). However, several of the putative UPRmt factors (e.g. ATP-dependent Clp protease proteolytic subunit (*Clpp*), DnaJ homolog subfamily A member 3 - mitochondrial (*Dnaj3a3*), endonuclease G (*Endog*), NADH dehydrogenase ubiquinone 1 beta subcomplex subunit 2 (*Ndufb2*), mitochondrial-processing peptidase subunit beta (*Pmpcb*), and ATP-

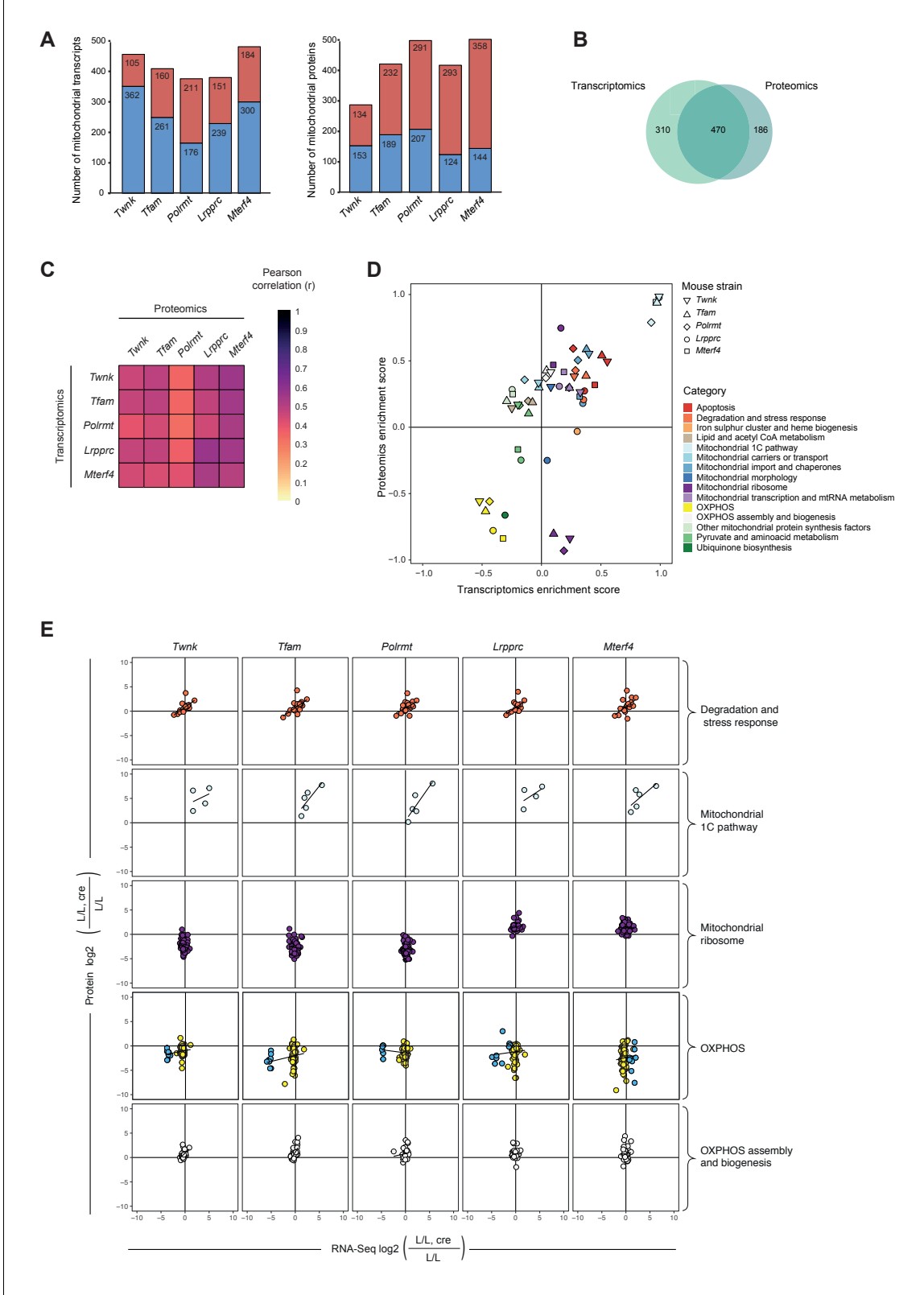

**Figure 4.** Remodeling of the mitochondrial transcriptome and proteome upon severe mitochondrial dysfunction. Files are provided in *Supplementary files 1* and *4*. (A) Number of significantly changed transcripts (left) and mitochondrial proteins (right) quantified from L/L, cre compared to L/L; red: increased, blue: decreased. (B) Venn diagram of number of mitochondrial transcripts and proteins quantified and significant in ≥1 knockout strain. (C) Correlations of the L/L, cre versus L/L fold changes of significantly regulated mitochondrial transcripts and proteins. (D) Mitochondrial
*Figure 4 continued on next page*

*Figure 4 continued*

transcriptomic and proteomic 2D enrichment analysis showing the trend and degree of regulation of 15 functional categories in all different knockouts.
(E) Scatterplots plots of the L/L, cre versus L/L fold changes of mitochondrial transcripts and proteins in different knockouts in a selection of categories. Black line indicates the trend. Same color code applied as in *Figure 4D* except for the OXPHOS category where the mitochondrial-encoded genes are colored in blue and the nuclear-encoded genes are colored in yellow.

DOI: https://doi.org/10.7554/eLife.30952.016

The following source data and figure supplements are available for figure 4:

**Source data 1.** Pearson correlation coefficient matrixes of fold changes of L/L, cre versus L/L of significantly regulated mitochondrial transcripts and proteins.

DOI: https://doi.org/10.7554/eLife.30952.020

**Figure supplement 1.** Most of the identified transcripts of nuclear genes encoding mitochondrial proteins are decreased, whereas most of the quantified mitochondrial proteins were increased in abundance.

DOI: https://doi.org/10.7554/eLife.30952.017

**Figure supplement 2.** Correlations of the L/L, cre versus L/L fold changes of significantly regulated mitochondrial transcripts and proteins.

DOI: https://doi.org/10.7554/eLife.30952.018

**Figure supplement 3.** Distribution of MYC and ATF4 target genes in scatterplots plots of the L/L, cre versus L/L fold changes of mitochondrial transcripts and proteins in different knockouts in a selection of categories.

DOI: https://doi.org/10.7554/eLife.30952.019

dependent zinc metalloprotease (*Yme1l1*)) did not present such an increase at the transcripts level in the five knockouts. This finding argues against the existence of a transcriptional program that regulates the expression of these factors upon mitochondrial stress. Contrary to its homolog in worm, the mammalian mitochondrial matrix protease CLPP was recently reported to neither regulate nor be required for UPRmt-like response in mouse heart (*Seiferling et al., 2016*). Supporting this finding, we detected no change of CLPP protein levels in our knockout mouse models (box plots in *Supplementary file 8*).

## Mitochondrial 1C pathway enzymes are up-regulated early in the progression of OXPHOS dysfunction

We found a markedly strong increase of the mitochondrial portion of the 1C pathway in all knockout mouse strains, both at the transcript and protein level (*Figure 6A,B*). In contrast, the cytosolic C-1-tetrahydrofolate synthase (MTHFD1) enzyme was unaltered at the protein level (*Figure 6C*). Next, we applied mass spectrometry analyses for targeted metabolomics and found increased levels of the 1C donors glycine, serine and sarcosine, consistent with a general up-regulation of the mitochondrial 1C pathway (*Figure 6D*). Temporal proteomic analysis of *Lrpprc* knockout mice further demonstrated that the levels of MTHFD2 are markedly increased (16-fold) early in the progression of OXPHOS dysfunction (*Figure 6E*). The up-regulation of the mitochondrial 1C pathway is thus an early event in the development of mitochondrial dysfunction. Our findings support a link between mitochondrial dysfunction and the 1C pathway (*Bao et al., 2016*; *Nikkanen et al., 2016*), which supplies 1C units for essential cellular processes, such as methylation, purine de novo synthesis and nicotinamide adenine dinucleotide phosphate (NADPH) synthesis (*Ducker and Rabinowitz, 2017*).

## Increased proline synthesis from glutamate upon OXPHOS dysfunction

ALDH18A1 was among the most up-regulated proteins (>1000 fold increase) in most knockout mouse strains (*Figure 7A,B*; *Figure 4—figure supplement 1B*). Moreover, the temporal proteomic analysis showed a steep increase of ALDH18A1 levels (*Figure 7C*) detectable early in the progression of OXPHOS dysfunction (*Mourier et al., 2014*). Because ALDH18A1 is the first enzyme of the glutamate to proline conversion pathway (*Pérez-Arellano et al., 2010*), we also investigated the other enzymes of this pathway. There was a substantial increase of the protein levels of PYCR1 and PYCR2 (*Figure 7A,B*; *Figure 4—figure supplement 1B*), whereas the delta-1-pyrroline-5-carboxylate dehydrogenase (ALDH4A1) and proline dehydrogenase 1 (PRODH), which catalyze the reverse reaction from proline to glutamate (*Bender et al., 2005*), were normal (*Figure 7B*). The levels of transcripts showed a similar trend as the levels of proteins in the glutamate to proline conversion pathway, with increased levels of *Aldh18a1* and *Pycr1,* and decreased levels of *Aldh4a1* transcripts (*Figure 7B*). To directly assess the effects generated by these protein expression changes, we

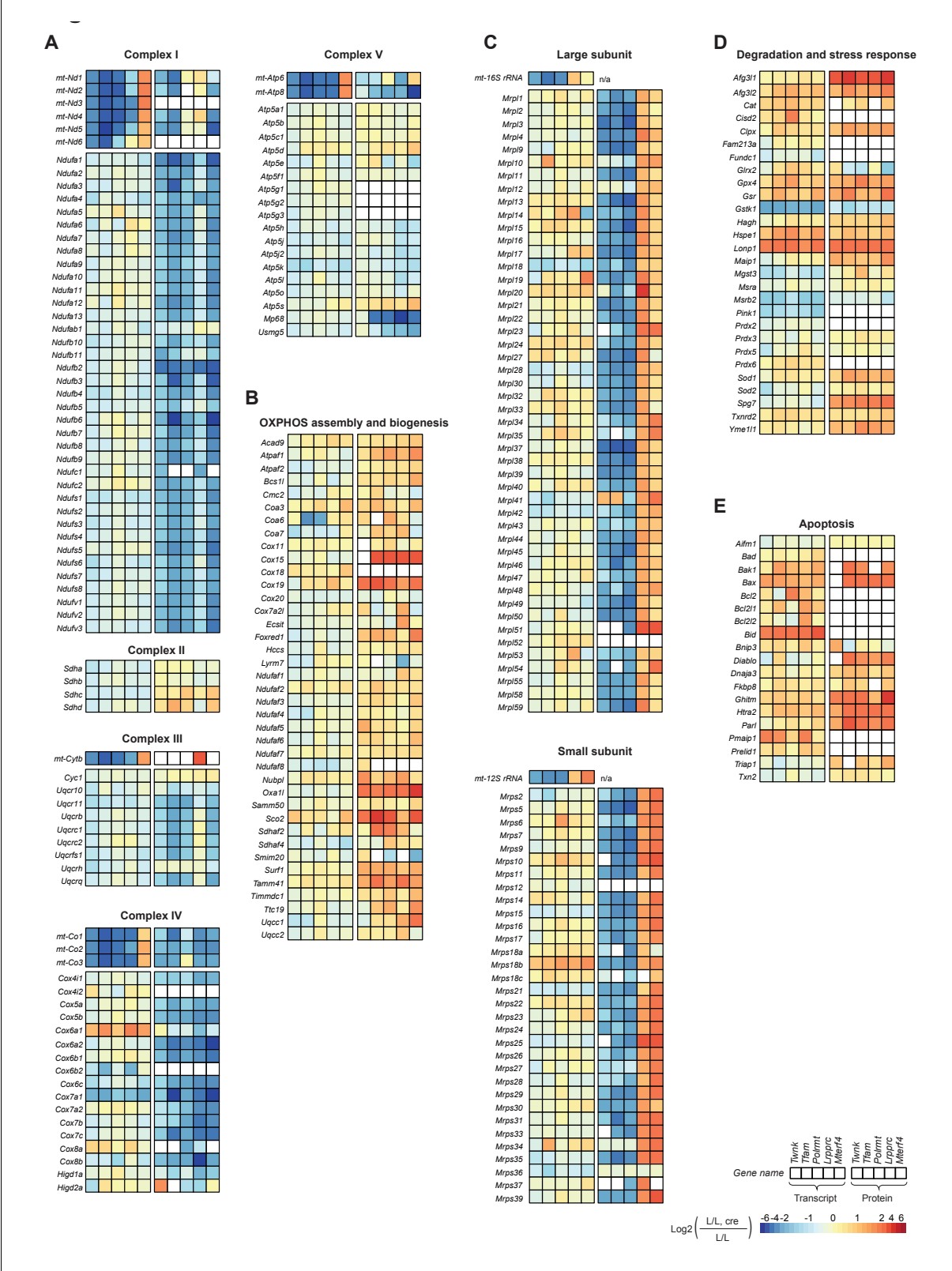

**Figure 5.** Effects of impaired mtDNA gene expression on mitochondrial protein levels in heart. Heatmaps illustrating the fold-change transcript (left) and mitochondrial protein (right) levels in L/L, cre and L/L mouse hearts in alphabetical order; blank boxes: not detected or not quantified. Adjusted p<0.05 in≥1 knockout strain. (**A**) OXPHOS complexes (complex II is only nuclear encoded). (**B**) OXPHOS assembly. (**C**) Mitochondrial ribosomal

*Figure 5 continued on next page*

*Figure 5 continued*

proteins; n/a = not applicable. (**D**) Degradation and stress response. (**E**) Apoptosis. For a selection of mitochondrial proteins the steady state levels were verified by immunoblotting shown in *Figure 5—figure supplement 2*.

DOI: https://doi.org/10.7554/eLife.30952.021

The following figure supplements are available for figure 5:

**Figure supplement 1.** Proteins regulating mitochondrial morphology, iron sulphur cluster and heme biogenesis are increased.

DOI: https://doi.org/10.7554/eLife.30952.022

**Figure supplement 2.** Immunoblot of several mitochondrial proteins in extracts from L/L, cre and L/L hearts; Loading of 6 different membranes: SDHA, SDHB or VDAC1.

DOI: https://doi.org/10.7554/eLife.30952.023

determined the glutamate and proline levels by targeted metabolomics and found a drastic increase in the proline to glutamate ratio in the knockout mouse hearts (*Figure 7D*). A possible consequence of this alteration is that tricarboxylic cycle intermediates may decrease, as α-ketoglutaric acid is converted to glutamine, which thereafter undergoes further degradation to proline.

## Secondary coenzyme Q deficiency develops in response to OXPHOS dysfunction

Very surprisingly, we detected that the levels of most of the intra-mitochondrial enzymes necessary for Q biosynthesis were severely reduced in all knockouts (*Figure 8A,B*). Q is an essential electron shuttle in the mitochondrial respiratory chain (*Milenkovic et al., 2017*; *Wang and Hekimi, 2016*) and we have previously reported that knockout of mitofusin 2 (*Mfn2*), a mitochondrial outer membrane protein involved in mitochondrial fusion (*Rojo et al., 2002*) and tethering of mitochondria to the endoplasmic reticulum (*de Brito and Scorrano, 2008*), causes Q deficiency by down-regulation of the cytosolic mevalonate (terpenoid/isoprenoid) pathway (*Mourier et al., 2015*). However, all of the knockout strains studied here had normal or slightly increased levels of mevalonate pathway enzymes such as HMG-CoA synthase, (HMGCS1) and farnesyl pyrophosphate synthase (FDPS; *Figure 8C*). The levels of transcripts encoding cytoplasmic HMGCS1, 3-hydrox-3-ymethylglutaryl-CoA reductase (HMGCR) and FDPS were either unaffected or increased in all knockouts (*Figure 8D*; *Figure 8—figure supplement 1A*), indicating that the mevalonate pathway is not impaired when mtDNA gene expression is disrupted.

The decreased levels of enzymes necessary for intra-mitochondrial Q biosynthesis were apparent at an early stage in the *Lrpprc* knockout strain and there was an accompanying decline in levels of the COQ3, COQ5, COQ6, COQ7, COQ8A, COQ9 and COQ10A enzymes as the OXPHOS deficiency progressed further (*Figure 8E,F*). In contrast, the protein levels of two of the Q biosynthesis enzymes, PDSS2 and COQ8A, increased as the OXPHOS deficiency progressed (*Figure 8F*). The decreased levels of enzymes of the mitochondrial Q biosynthesis pathway were likely due to impaired protein stability or targeted protein degradation as the levels of transcripts from most of the corresponding genes were unaltered or moderately reduced. However, the atypical kinases COQ8A and COQ8B showed strong opposite responses (*Figure 8D*), consistent with the observed differences at the protein level (*Figure 8B,F*). Interestingly, COQ8A and COQ8B have recently been suggested to interact differentially with the Q biosynthesis complex depending on whether metabolism is adapting to glycolytic or respiratory conditions (*Floyd et al., 2016*).

Finally, we assessed whether the decreased levels of enzymes necessary for the intra-mitochondrial Q biosynthesis directly affected the cellular steady-state levels of Q. In mice, Q9 is much more abundant than Q10 (*Tang et al., 2004*) and we therefore measured the levels of both compounds in mouse heart tissue homogenates using targeted mass spectrometry. In control hearts, Q9 was around tenfold more abundant than Q10 (*Figure 8G*), whereas the knockouts showed a profound decrease of Q9, consistent with the observed down-regulation of the intra-mitochondrial Q biosynthesis pathway.

## Discussion and conclusions

Here, we present a systematic comparative analysis of the mitoproteome and the global transcriptome of five knockout mouse strains with disruption of key genes for mtDNA gene expression in the

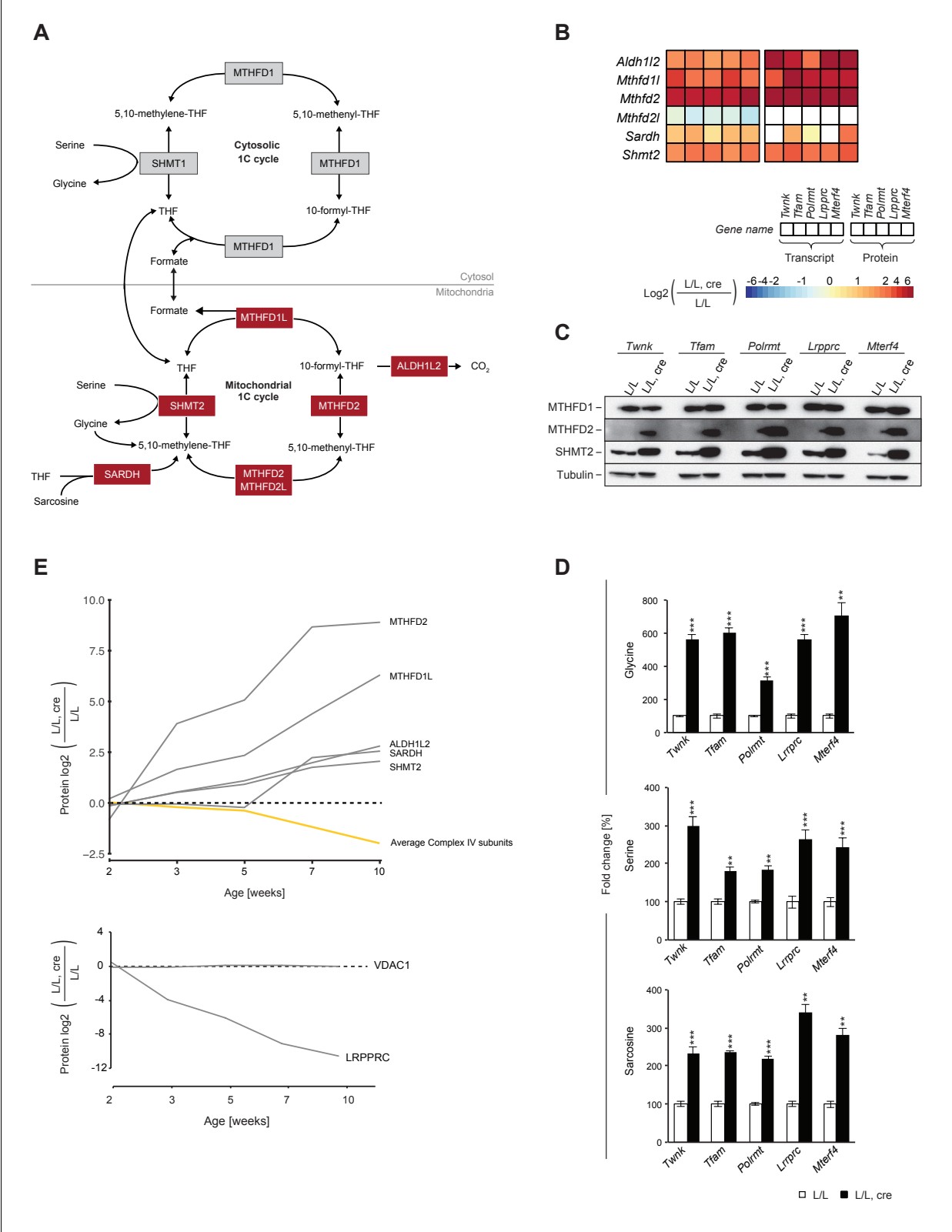

**Figure 6.** Up-regulation of the enzymes of the mitochondrial 1C pathway happens before deficient OXPHOS is detectable in mouse heart. (**A**) Scheme of 1C pathway. Colored boxes: protein levels; red: increased, grey: not detected or not quantified. (**B**) Heatmaps showing the fold-change in transcript (left) and protein (right) levels in alphabetical order of L/L, cre and L/L mouse hearts of the 1C pathway; p<0.0001 in≥1 knockout strain. (**C**) Immunoblot of enzymes of the 1C pathway in total protein extracts from L/L, cre and L/L; Loading: tubulin. (**D**) Quantification of 1C donor metabolite levels in L/L,

*Figure 6 continued on next page*

*Figure 6 continued*

cre and L/L. Graphs represent mean ± SEM (*p<0.05, **p<0.01, ***p<0.001). (E) Time point analysis of protein levels of enzymes of the 1C pathway (top), and LRPPRC and VDAC (bottom) in *Lrpprc* knockout hearts compared to controls. Yellow line: average value of nuclear and mitochondrial encoded OXPHOS complex IV subunits. Adjusted p<0.05, except for VDAC. LRPPRC and VDAC protein levels at the different time points were verified by immunoblotting presented in *Figure 6—figure supplement 1*.

DOI: https://doi.org/10.7554/eLife.30952.024

The following source data and figure supplement are available for figure 6:

**Source data 1.** Determination of 1C pathway donor metabolites.

DOI: https://doi.org/10.7554/eLife.30952.025

**Figure supplement 1.** Steady-state LRPPRC protein levels at different time points in mitochondrial extracts from *Lrpprc* L/L, cre and L/L hearts; Loading: VDAC1.

DOI: https://doi.org/10.7554/eLife.30952.026

heart. Interestingly, the mitoproteome in mouse heart is mainly regulated at the protein level in response to severe mitochondrial dysfunction. Even though both, the mitochondrial transcriptome and mitoproteome, are strongly affected, the changes in the mitoproteome do not mirror the

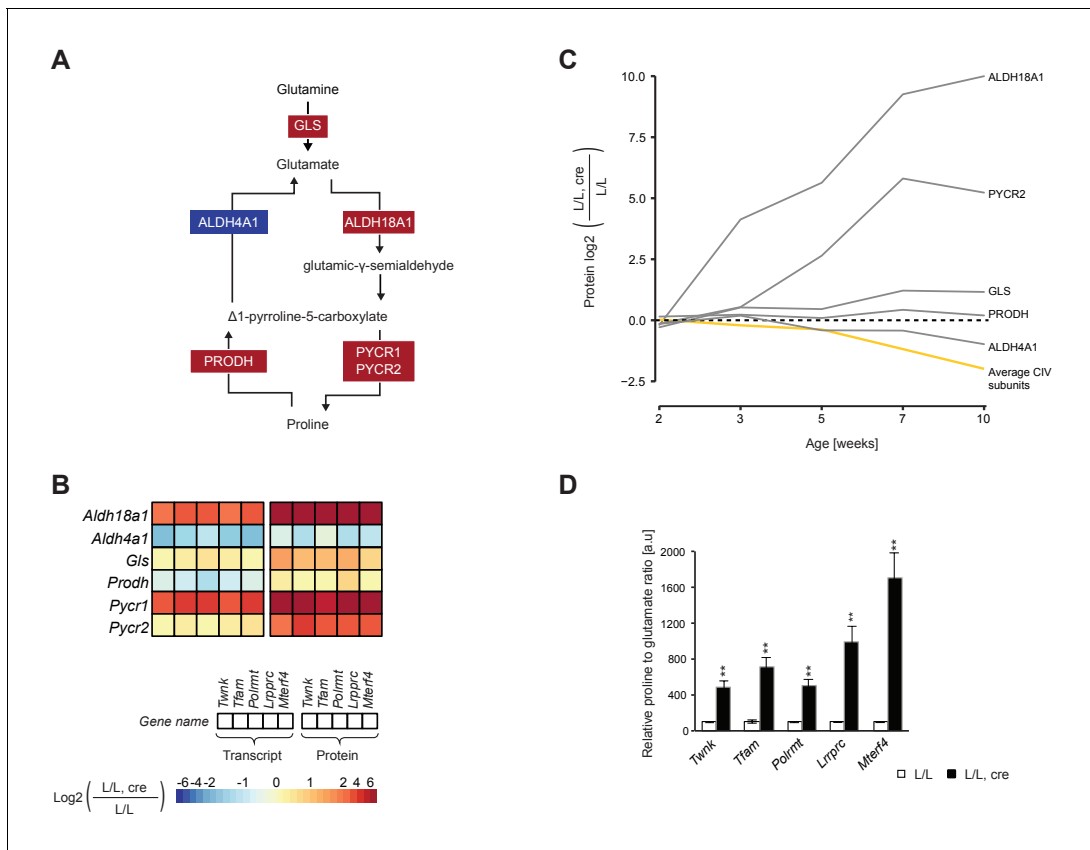

**Figure 7.** Up-regulated glutamate to proline conversion in mitochondrial OXPHOS deficient heart. (A) Scheme of the glutamate to proline conversion pathway. Colored boxes: protein levels; red: increased, blue: decreased. (B) Heatmaps illustrating the fold-change transcript (left) and protein (right) levels. Adjusted p<0.05 in≥1 knockout strain for genes encoding mitochondrial transcript or protein levels. (C) Time point analysis of protein levels of enzymes of the glutamate to proline conversion pathway in *Lrpprc* knockout hearts compared to controls. Yellow line: average value of nuclear and mitochondrial encoded OXPHOS complex IV subunits. Adjusted p<0.05. (D) Quantification of proline and glutamate in different L/L, cre and L/L mouse hearts. Error bars:± SEM; **p<0.01; two-tailed unpaired Student's t-test.

DOI: https://doi.org/10.7554/eLife.30952.027

The following source data is available for figure 7:

**Source data 1.** Determination of proline and glutamate metabolites.

DOI: https://doi.org/10.7554/eLife.30952.028

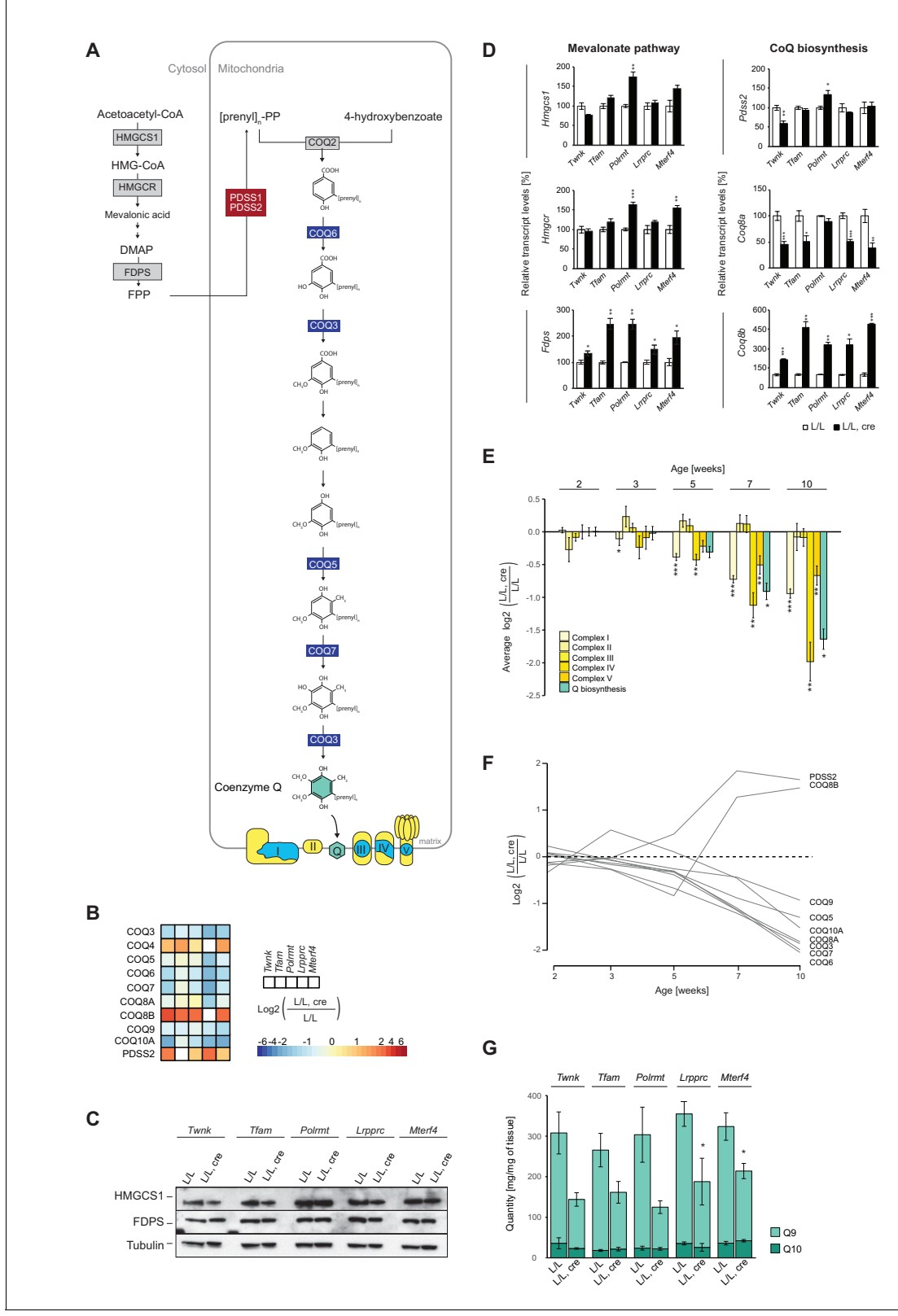

**Figure 8.** OXPHOS dysfunction leads to decreased cellular Q levels, but the enzymes of the mevalonate pathway are normal. (**A**) Scheme of the mevalonate and Q biosynthesis pathways, and OXPHOS complexes (Nuclear-encoded OXPHOS proteins are shown in yellow and mtDNA-encoded OXPHOS proteins in blue). Colored boxes: protein levels; red: increased, blue: decreased, grey: not detected or not quantified. (**B**) Heatmaps illustrating the fold-change protein levels of the Q biosynthesis pathway in alphabetical order of L/L, cre and L/L mouse hearts; blank boxes: not

*Figure 8 continued on next page*

*Figure 8 continued*

detected or not quantified proteins; p<0.05 in≥1 knockout strain. (C) Immunoblot of enzymes of the mevalonate pathway on total protein extracts from different L/L, cre and L/L hearts. Loading: tubulin. (D) Transcript levels of genes encoding enzymes of the mevalonate and coenzyme Q synthesis pathway in L/L, cre and L/L hearts. Normalization: B2M (beta-2-microglobulin). (E) Protein levels of OXPHOS complexes I-V and the downregulated Q biosynthesis enzymes at different time points in *Lrpprc* knockout mouse hearts compared to controls. The graph represents a mean log2 fold-change of all the proteins in that category. (F) Time point analysis of protein levels of enzymes of the Q biosynthesis pathway in *Lrpprc* knockout mouse hearts compared to controls. Adjusted p across time <0.05. (G) Quinone quantification (**Q9 and Q10**) in different L/L, cre and L/L mouse hearts. Error bars:± SEM; *p<0.05, **p<0.01, ***p<0.001; two-tailed unpaired Student's t-test.

DOI: https://doi.org/10.7554/eLife.30952.029

The following source data and figure supplement are available for figure 8:

**Source data 1.** qRT-PCR of genes encoding ubiquinone and mevalonate pathway enzymes in the five knockout mouse strains.
DOI: https://doi.org/10.7554/eLife.30952.031
**Source data 2.** Determination of coenzyme Q9 and 10.
DOI: https://doi.org/10.7554/eLife.30952.032
**Figure supplement 1.** Transcript levels of genes encoding for enzymes of the mevalonate and the Q synthesis pathways.
DOI: https://doi.org/10.7554/eLife.30952.030

transcript levels. Thus, the majority of differentially expressed mitochondrial proteins are increased under severe mitochondrial dysfunction, due to posttranscriptional regulation and the exact mechanisms of those compensatory responses remain to be discovered.

The combination of knockout strains that were inter-compared on a transcriptomic and mitoproteomic level allowed us to detect common expression patterns, either of individual genes or of whole functional categories, and to determine whether these patterns depended on the level at which mtDNA gene expression was disrupted. To exemplify, reduced steady-state levels of mt-rRNAs lead to massively reduced levels of all detected mitoribosomal proteins, whereas increased mt-rRNA levels lead to increased protein levels. This shows that the mtDNA-encoded rRNAs are essential for the stability of the nucleus-encoded mitoribosomal proteins and mitoribosome assembly. Our finding is further supported by a recent study where loss of mature mt-rRNAs impairs mitoribosome assembly in an individual mouse strain with disruption of a factor required for polycistronic mt-RNA processing (*Rackham et al., 2016*). Interestingly, we found the same pattern for several genes annotated to be factors of mitochondrial transcription and mt-RNA metabolism (*Rars2*, *Vars2*, and *Trmu*) or mitochondrial protein synthesis (*Mtg1* and *Mtif3*). It is tempting to speculate that those genes encode proteins that either require a functional mitoribosome for their stability or are also mitoribosomal proteins even though some of them have not yet been defined as such. The mitochondrial ribosome-associated GTPase 1 (MTG1) has been reported to associate with the large subunit of the mitoribosome (*Kotani et al., 2013*) and the mitochondrial translation initiation factor IF-3 (MTIF3) binds to the 28S ribosomal subunit in cross-linking studies (*Koc and Spremulli, 2002*). On the contrary, *Mrps36* encoding the mitochondrial 28S ribosomal protein S36 is annotated as a component of the small mitochondrial ribosome subunit, but its expression pattern in our study rather suggests a different function. A recent study in bakers yeast reports that MRPS36 (KGDH) has a conserved role in the organization of mitochondrial α-ketoglutarate dehydrogenase complexes and suggests that it is not a subunit of the mitochondrial ribosome (*Heublein et al., 2014*).

The resources we present here also exemplify the myriad of altered cellular processes that accompany OXPHOS deficiency and that could contribute to the pathogenic processes leading to mitochondrial disease. The enrichment of transcripts of genes of the eIF2 signaling pathway, with ATF4 and MYC amongst the top regulators, is consistent with a cellular response to stress, whereby transcription of stress response genes mediated by transcription factors, such as ATF4, ATF5 and MYC, is promoted (*Holcik and Sonenberg, 2005*). Different stress responses activated by mitochondrial dysfunction have been described, including up-regulation of mitochondrial biogenesis, increase of mitochondrial proteases and chaperones, and changes in energy metabolism (*Hansson et al., 2004*). It was recently shown that mitochondrial myopathy mouse models with mtDNA replication disorders have increased protein levels of the mitochondrial 1C pathway enzymes MTHFD2 and MTHFD1L, as well as increased transcript levels of the genes encoding for key enzymes of the serine synthesis pathway (*Nikkanen et al., 2016*). When mtDNA is depleted, this effect was shown to be mediated by ATF4 (*Bao et al., 2016*) and to be under the control of mammalian target of rapamycin

complex 1 (mTORC1) in mouse skeletal muscle (*Khan et al., 2017*). A model has been proposed whereby the impaired OXPHOS activity decreases formate levels and up-regulation of the mitochondrial 1C pathway occurs as an adaptive response (*Ducker and Rabinowitz, 2017*). Our proteomic and transcriptomic studies confirm an up-regulation of the enzymes of the mitochondrial 1C pathway as well as an increase of 1C donors, and further show that the increase of the 1C enzyme MTHFD2 is already detectable at an early stage of the progression of OXPHOS deficiency. Given the high expression of *Mthfd2* at the transcript and protein level in early stages of OXPHOS deficiency, our study suggests *Mthfd2* as a marker for early stages of mitochondrial dysfunction. Our results support the role of ATF4 in modulating the transcriptional response to mitochondrial dysfunction and are in line with the early expression of *Atf4* in a mt-tRNA synthetase (*Dars2*) deficient knockout mouse (*Dogan et al., 2014*). Notably, the upregulation of 1C metabolism and target genes of ATF4 is a common response in all of the knockout mouse strains regardless of the stage where mtDNA gene expression is disrupted, indicating that it is a general response to loss of mtDNA gene expression or OXPHOS dysfunction rather than triggered exclusively by mtDNA replication defects or by direct sensing of components of the mtDNA gene expression system (*Bao et al., 2016*; *Nikkanen et al., 2016*). Moreover, our data supports the model whereby an integrated stress response program mediated by ATF4 is activated upon mitochondrial dysfunction in mammals (*Khan et al., 2017*; *Seiferling et al., 2016*). A central element of the proposed integrated stress response is the release of metabolic signaling proteins such as the fibroblast growth factor 21 (FGF21), which is secreted after OXPHOS deficiency and can affect whole-body metabolism (*Dogan et al., 2014*). Since the knockout models used in this study are heart and skeletal muscle specific knockouts, the possibility that some of the metabolic alterations we detect in heart is due to signaling molecules released from skeletal muscle cannot be excluded.

MYC is an important regulator of cellular proliferation and tissue homeostasis (*Kress et al., 2015*). In addition, studies in cell lines have identified more than 400 nuclear genes encoding mitochondrial proteins as potential MYC targets, including proteins involved in mitochondrial protein import, OXPHOS assembly, mitoribosome and mtDNA gene expression (*Morrish and Hockenbery, 2014*). Therefore, it is possible that MYC contributes to the metabolic rewiring that is caused by mitochondrial dysfunction. MYC is widely known as an oncogene (*Meyer and Penn, 2008*) and the expression of genes encoding mitoribosomal proteins, mitoribosome assembly factors and mitochondrial translation factors is modified in numerous cancers (*Kim et al., 2017*). Remarkably, all knockouts in our study show a drastic increase of *Myc* transcripts and significant changes in several MYC target genes, which supports the idea that MYC is involved in remodeling mitochondrial metabolism under severe mitochondrial dysfunction. In line with this hypothesis, we found that the proline biosynthesis from glutamate was promoted upon mitochondrial dysfunction in all analyzed mouse strains. An important role of MYC in this process was previously reported in human cancer cell lines, where MYC inhibits *PRODH* and promotes the transcription of *PYCR1* and *ALDH18A1* (*Liu et al., 2012*). It should be noted that induction of *Myc* in the myocardium of wild-type mice was shown to increase hypertrophic growth and mitochondrial biogenesis in response to pathological stressors like ischemia/reperfusion (*Ahuja et al., 2010*). Thus, the tissue specificity of MYC on remodeling mitochondrial biogenesis and metabolism should be considered. Furthermore, our data suggest that proline levels should be evaluated as a potential marker for mitochondrial deficiency in patients. Importantly, *Pycr1* and *Aldh18a1* are target genes for both ATF4 and CHOP (*Han et al., 2013*) and therefore the effect we observed could be mediated by either of these factors or by MYC. Inhibition of mTORC1 with rapamycin was recently demonstrated to slow the progression of mitochondrial myopathy in mouse models by improving several hallmarks of mitochondrial dysfunction, resulting in reduced expression of *Atf4, Fgf21*, mitochondrial chaperones and key enzymes of the 1C pathway (*Khan et al., 2017*). The identification and characterization of signaling pathways and secondary responses that contribute to the pathophysiology of mitochondrial diseases may allow specific interventions to treat mitochondrial diseases.

Remarkably, we found that impaired mtDNA gene expression in mouse heart causes secondary Q deficiency as a result of defective intra-mitochondrial Q biosynthesis and not via the cytosolic mevalonate pathway. Moreover, we analyzed the sequential changes of the protein levels of the Q biosynthesis pathway, and provide evidence that COQ8A and COQ8B have key regulatory roles in adapting mitochondrial function and Q biosynthesis to changes in metabolic requirements in vivo. The general downregulation of the intramitochondrial Q biosynthesis pathway could possibly serve

to adjust Q levels to the available respiratory chain enzymes. In addition to its function in electron transfer, Q is a ubiquitous antioxidant for lipids present in all the membranes within the cell (*Wang and Hekimi, 2016*). However, we propose that a more likely explanation is that the loss of OXPHOS complexes affects the integrity of the inner mitochondrial membrane thereby destabilizing the Q biosynthesis complex. It has previously been observed that secondary changes may contribute to disease progression, for example we have recently reported that a mouse model with male infertility caused by mitochondrial dysfunction has unfavorable pathophysiological responses (*Jiang et al., 2017*). It is thus possible that treatments interfering with secondary responses, for example Q deficiency, may be a valid treatment strategy.

The connection between OXPHOS dysfunction and reduced Q synthesis that we describe here is of fundamental importance and has implications for several medical areas. It should be noted that inherited, autosomal recessive forms of Q deficiency in patients respond well to oral Q supplementation, as documented in several human and animal studies (*Garrido-Maraver et al., 2014*; *Wang and Hekimi, 2016*). Furthermore, Q is widely used to treat patients with different types of mitochondrial diseases although the rationale for such a treatment is much debated (*Chinnery et al., 2006*). Our data provide experimental support for the hypothesis that the large group of patients with disease caused by impaired mtDNA gene expression, due to mutations in either mtDNA or nuclear DNA, may develop secondary Q deficiency. Q supplementation may thus improve electron transport capacity and we therefore propose that Q measurements should be performed in all patients with mitochondrial disease as Q supplementation could be a beneficial treatment option. Notably, mitochondrial function is known to decline during mammalian aging (*Greaves et al., 2012*) and one important explanation for this decrease is accumulation of somatic mtDNA mutations that impair mtDNA gene expression (*Larsson, 2010*).

The function and structure of mitochondria have been the subject of intensive research since the discovery of these organelles. Yet, the investigation of mitochondrial diseases with novel techniques has revealed that the underlying pathogenic processes are much more complex than previously expected and are intricately connected to a wide range of cellular functions. Here we provide a high quality list of gene expression changes at the RNA and protein level in a series of mouse mutants with disruption of mtDNA expression. These mouse models all develop a severe OXPHOS deficiency in heart with massive consequences for a range of cellular metabolic pathways. Importantly, some of these secondary changes, such as the Q deficiency we describe here, may be possible to target therapeutically. The data resources we present here generated from in vivo sources will not only open new avenues for research, but also provide a rich source for information of value for patient diagnosis and research on future treatment strategies.

# Materials and methods

## Key resources table

| Reagent type (species) or resource | Designation | Source or reference | Identifiers | Additional information |
|---|---|---|---|---|
| Organism, C57BL/6N (*Mus musculus*) | *Twnk*$^{LoxP/LoxP}$; Twnk L/L | *Milenkovic et al., 2013* | RRID: MGI:5496889 | |
| Organism, C57BL/6N (*Mus musculus*) | *Twnk*$^{LoxP/LoxP}$, +/Ckmm-cre; Twnk L/L, cre | *Milenkovic et al., 2013* | RRID: MGI:5496891 | |
| Organism, C57BL/6N (*Mus musculus*) | *Tfam*$^{LoxP/LoxP}$; Tfam L/L | *Larsson et al., 1998* | RRID: MGI:2177633 | |
| Organism, C57BL/6N (*Mus musculus*) | *Tfam*$^{LoxP/LoxP}$, +/Ckmm-cre; Tfam L/L, cre | *Larsson et al., 1998* | RRID: MGI:2177634 | |
| Organism, C57BL/6N (*Mus musculus*) | *Polrmt*$^{LoxP/LoxP}$; Polrmt L/L | *Kühl et al., 2014, 2016* | MGI:5704129 | |
| Organism, C57BL/6N (*Mus musculus*) | *Polrmt*$^{LoxP/LoxP}$, +/Ckmm-cre; Polrmt L/L, cre | *Kühl et al., 2014, 2016* | RRID: MGI:5704131 | |
| Organism, C57BL/6N (*Mus musculus*) | *Lrpprc*$^{LoxP/LoxP}$; Lrpprc L/L | *Ruzzenente et al., 2012* | RRID: MGI:5438915 | |

*Continued on next page*

*Continued*

| Reagent type (species) or resource | Designation | Source or reference | Identifiers | Additional information |
|---|---|---|---|---|
| Organism, C57BL/6N (*Mus musculus*) | *Lrpprc*<sup>LoxP/LoxP</sup>, +/*Ckmm-cre*; *Lrpprc* L/L, cre | *Ruzzenente et al., 2012* | RRID: MGI:5438914 | |
| Organism, C57BL/6N (*Mus musculus*) | *Mterf4*<sup>LoxP/LoxP</sup>; *Mterf4* L/L | *Cámara et al., 2011* | RRID: MGI:5288508 | |
| Organism, C57BL/6N (*Mus musculus*) | *Mterf4*<sup>LoxP/LoxP</sup>, +/*Ckmm-cre*; *Mterf4* L/L, cre | *Cámara et al., 2011* | RRID: MGI:5292478 | |
| Antibody | ALDH18A1 | Thermofisher Scientific | Cat#PA5-19392 RRID: AB_10985670 | (1:200) |
| Antibody | CLPP | Sigma-Aldrich | Cat#WH0008192M1 RRID: AB_1840782 | (1:300) |
| Antibody | COX4 | Cell Signaling | Cat#4850 RRID: AB_2085424 | (1:500) |
| Antibody | CS | Abcam | Cat#ab129095 RRID: AB_11143209 | (1:200) |
| Antibody | FDPS | Abcam | Cat#ab189874 RRID: AB_2716301 | (1:500) |
| Antibody | GLS | Abcam | Cat#ab93434 RRID: AB_10561964 | (1:200) |
| Antibody | HMGCS1 | Abcam | Cat#ab194971 RRID: AB_2716299 | (1:500) |
| Antibody | HSPA9/mtHSP70/Grp75 | Abcam | Cat#ab82591 RRID: AB_1860633 | (1:200) |
| Antibody | LONP1 | Abcam | Cat#ab103809 RRID: AB_10858161 | (1:500) |
| Antibody | LRPPRC mouse | N.-G. Larsson; *Ruzzenente et al., 2012* | RRID: AB_2716302 | (1:1000) |
| Antibody | MRPL37 | Sigma-Aldrich | Cat#HPA025826 RRID: AB_1854106 | (1:500) |
| Antibody | MRLP44 | Proteintech | Cat#16394–1-AP RRID: AB_2146062 | (1:300) |
| Antibody | MRPS35 | Proteintech | Cat#16457–1-AP RRID: AB_2146521 | (1:500) |
| Antibody | MTHFD1 | Abcam | Cat#ab103698 RRID: AB_10862775 | (1:500) |
| Antibody | MTHFD2 | Abcam | Cat#ab37840 RRID: AB_776544 | (1:500) |
| Antibody | NDUFA9 | Abcam | Cat#ab14713 RRID: AB_301431 | (1:500) |
| Antibody | POLRMT mouse | N.-G. Larsson; *Kühl et al., 2014* | RRID: AB_2716297 | |
| Antibody | PYCR1 | Proteintech | Cat#13108–1-AP RRID: AB_2174878 | (1:200) |
| Antibody | SDHA | Thermofisher Scientific | Cat#459200 RRID: AB_2532231 | (1:100) |
| Antibody | SHMT2 | Sigma-Aldrich | Cat#HPA020543 RRID: AB_1856833 | (1:500) |
| Antibody | TFAM | Abcam | Cat#ab131607 RRID: AB_11154693 | (1:500) |
| Antibody | Total OXPHOS Rodent WB Antibody Cocktail | Abcam | Cat#ab110413 RRID: AB_2629281 | (1:1000) |
| Antibody | Tubulin | Cell Signaling | Cat#2125 RRID: AB_2619646 | (1:1000) |

*Continued on next page*

*Continued*

| Reagent type (species) or resource | Designation | Source or reference | Identifiers | Additional information |
|---|---|---|---|---|
| Antibody | TWINKLE mouse | N.-G. Larsson, *Milenkovic et al., 2013* | RRID: AB_2716298 | |
| Antibody | UQCRFS1 | Abcam | Cat#ab131152 RRID:AB_2716303 | (1:200) |
| Antibody | VDAC1 | Millipore | Cat#MABN504 RRID:AB_2716304 | (1:1000) |
| Sequence-based reagent | Taqman Assay - Mouse *Adck3* | Life technologies | Mm00469737_m1 | |
| Sequence-based reagent | Taqman Assay - Mouse *Adck4* | Life technologies | Mm00505363_m1 | |
| Sequence-based reagent | Taqman Assay - Mouse *Atf4* | Life technologies | Mm00515325_m1 | |
| Sequence-based reagent | Taqman Assay - Mouse *Myc* | Life technologies | Mm00487804_m1 | |
| Sequence-based reagent | Taqman Assay - Mouse *Coq2* | Life technologies | Mm01203260_g1 | |
| Sequence-based reagent | Taqman Assay - Mouse *Coq4* | Life technologies | Mm00618552_m1 | |
| Sequence-based reagent | Taqman Assay - Mouse *Coq5* | Life technologies | Mm00518239_m1 | |
| Sequence-based reagent | Taqman Assay - Mouse *Coq7* | Life technologies | Mm00501587_m1 | |
| Sequence-based reagent | Taqman Assay - Mouse *Fdps* | Life technologies | Mm00836315_g1 | |
| Sequence-based reagent | Taqman Assay - Mouse *Gabpa* | Life technologies | Mm00484598_m1 | |
| Sequence-based reagent | Taqman Assay - Mouse *Hmgcs1* | Life technologies | Mm01304569_m1 | |
| Sequence-based reagent | Taqman Assay - Mouse *Hmgcr* | Life technologies | Mm01282499_m1 | |
| Sequence-based reagent | Taqman Assay - Mouse *Mterf4* | Life technologies | Mm00508298_m1 | |
| Sequence-based reagent | Taqman Assay -Mouse *Nrf1* | Life technologies | Mm00447996_m1 | |
| Sequence-based reagent | Taqman Assay - Mouse *Pdss1* | Life technologies | Mm00450958_m1 | |
| Sequence-based reagent | Taqman Assay - Mouse *Pdss2* | Life technologies | Mm01190168_m1 | |
| Sequence-based reagent | Taqman Assay - Mouse *Ppargc1* | Life technologies | Mm00447183_m1 | |
| Sequence-based reagent | Taqman Assay - Mouse *Polrmt* | Life technologies | Mm00553272_m1 | |
| Sequence-based reagent | Taqman Assay - Mouse *Tfam* | Life technologies | Mm00627275_m1 | |
| Sequence-based reagent | Taqman Assay - Mouse *Peo1/Twnk* | Life technologies | Mm00467928_m1 | |
| Commercial assay or kit | miRNeasy Mini kit | Qiagen | Cat#217004 | |
| Commercial assay or kit | Ribo-Zero rRNA removal kit | Illumina | MRZH11124 | |
| Commercial assay or kit | Tru-Seq Sample preparation | Illumina | RS-122–2002 | |
| Commercial assay or kit | High Capacity cDNA revese transcription kit | Applied Biosystems | Cat#4368814 | |
| Commercial assay or kit | TaqMan Universal PCR Master Mix, No Amperase UNG | Applied Biosystems | Cat#4324020 | |
| Commercial assay or kit | Citrate Synthase Assay Kit | Sigma-Aldrich | Cat#CS0720 | |
| Chemical compound, drug | EDTA-free complete protease inhibitor cocktail | Roche | Cat#05056489001 | |
| Chemical compound, drug | PhosSTOP tablets | Roche | Cat#04906837001 | |
| Chemical compound, drug | Percoll | GE Heathcare | Cat#17-0891-02 | |
| Chemical compound, drug | Trypsin gold | Promega | Cat#V5280 | |
| Chemical compound, drug | Standard Coenzyme Q9 | Sigma-Aldrich | Cat#27597 | |
| Chemical compound, drug | Standard Coenzyme Q10 | Sigma-Aldrich | Cat#C9538 | |
| Chemical compound, drug | Standard Glutamate (Glutamic acid) | Sigma-Aldrich | Cat#G1251 | |
| Chemical compound, drug | Standard Glycine | Sigma-Aldrich | Cat#G7126 | |
| Chemical compound, drug | Standard Proline | Sigma-Aldrich | Cat#P0380 | |

*Continued on next page*

*Continued*

| Reagent type (species) or resource | Designation | Source or reference | Identifiers | Additional information |
|---|---|---|---|---|
| Chemical compound, drug | Standard Sarcosine | Sigma-Aldrich | Cat#S7672 | |
| Chemical compound, drug | Standard Serine | Sigma-Aldrich | Cat#S4500 | |
| Software, algorithm | Cytoscape v. 3.5.0 | *Shannon et al., 2003* | http://www.cytoscape.org RRID:SCR_003032 | |
| Software, algorithm | DESeq2 package R v. 3.3.2 | Other | *Love et al. (2014)* | |
| Software, algorithm | Ingenuity Pathway Analysis - Ingenuity Systems | Qiagen | www.ingenuity.com RRID:SCR_008653 | |
| Software, algorithm | iRegulon v. 1.3 | *Janky et al., 2014* | http://iregulon.aertslab.org/ download.html | |
| Software, algorithm | MaxQuant v. 1.5.2.8 | *Cox and Mann, 2008* | http://www.coxdocs.org/ doku.php id=maxquant:start RRID:SCR_014485 | |
| Software, algorithm | R - The R project for Statistical Computing | | https://www.r-project.org RRID:SCR_001905 | |
| Software, algorithm | Perseus | *Cox and Mann, 2012* | http://www.coxdocs.org/ doku.php?id=perseus:start | |
| Software, algorithm | TargetP v. 1.1 | *Emanuelsson et al., 2000*; *Nielsen et al., 1997* | http://www.cbs.dtu.dk/ services/TargetP/ | |
| Other | 1.9 mm ReproSil-Pur 120 C18-AQ media | Dr. Maisch | Cat#r119.aq | |
| Other | 25 cm, (75 mm internal diameter) PicoFrit analytical column | New Objective | Cat#PF7508250 | |

## Mouse work

Breeding of heart- and skeletal muscle-specific knockout strains for *Twnk* (RRID: MGI:5496891), *Tfam* (RRID: MGI:2177634), *Polrmt* (RRID: MGI:5704131), *Lrpprc* (RRID: MGI:5438914) and *Mterf4* (RRID: MGI:5292478) were performed as previously described in a C57Bl6/N background (*Cámara et al., 2011*; *Kühl et al., 2016*; *Larsson et al., 1998*; *Milenkovic et al., 2013*; *Ruzzenente et al., 2012*). For the comparison of the five knockout strains, mice were killed by cervical dislocation and mouse hearts were analyzed at the latest possible time point (*Twnk*, 12–16 weeks; *Tfam*, 8–9 weeks, *Polrmt*, 3–4 weeks; *Lrpprc*, 10–12 weeks, *Mterf4* 18–23 weeks). For the *Lrpprc* time course, mouse hearts were analyzed at 2, 3, 5, 7 and 10 weeks of age. Animals of both genders were used as available. Conditional knockouts (L/L, cre) and control (L/L) littermates from each knockout strain were randomly assigned to experimental groups to avoid any bias, whenever possible.

## Ultrapure mitochondria isolation, peptide digestion and clean up for label-free mass spectrometry

Mitochondria were isolated from mouse hearts using differential centrifugation as previously reported (*Kühl et al., 2016*). Crude mitochondrial pellets were washed in 1xM buffer (220 mM mannitol, 70 mM sucrose, 5 mM HEPES pH 7.4, 1 mM EGTA pH 7.4; pH was adjusted with potassium hydroxide; supplemented with EDTA-free complete protease inhibitor cocktail and PhosSTOP Tablets (Roche, Germany)) and purified on a Percoll density gradient (12%:19%:40% prepared with buffer 2xM) via centrifugation in a SW41 rotor at 15500 rpm at 4°C for 1 hr in a Beckman Coulter Optima L-100 XP ultracentrifuge using 14 mm ×89 mm Ultra-Clear Centrifuge Tubes (Beckman Instruments Inc., Germany). Mitochondria were harvested at the interphase between 19% and 40%, washed three times with buffer 1xM and mitochondrial pellets were frozen at −80°C. Purified frozen mitochondria pellets were suspended in lysis buffer (6 M guanidinium chloride, 10 mM Tris(2-carboxyethyl)phosphine hydrochloride, 40 mM chloroacetamide, and 100 mM Tris-HCl) (*Kulak et al., 2014*). After complete lysis, samples were diluted 1:10 in 20 mM Tris-HCL pH 8.0 and 80 µg of protein were mixed with 3 µg of Trypsin gold (Promega) and incubated overnight at 37°C to achieve

complete digestion. Peptides were cleaned with home-made STAGEtips (*Rappsilber et al., 2003*) (Empore Octadecyl C18; 3M, Germany) and eluted in 60% acetonitrile/0.1% formic acid buffer. Samples were dried in a SpeedVac apparatus (Eppendorf concentrator plus 5305) at 45°C and the peptides were suspended with 0.1% formic acid. Approximately 1.5 μg of peptides were analyzed by LC-MS/MS. All the samples of either the proteomic comparison of end-stage knockouts or of the temporal proteomic analysis of *Lrpprc* knockouts were prepared at the same time by the same person.

## LC-MS/MS analysis

For mass spectrometric analysis, peptides were separated on a 25 cm, 75 μm internal diameter Pico-Frit analytical column (New Objective, Part No. PF7508250) packed with 1.9 μm ReproSil-Pur 120 C18-AQ media (Dr. Maisch, Mat. No. r119.aq) using an EASY-nLC 1000 or EASY-nLC 1200 (Thermo Fisher Scientific, Germany). The column was maintained at 50°C. Buffer A and B were 0.1% formic acid in water and 0.1% formic acid in acetonitrile, respectively. For the analysis using the EASY-nLC 1200 system, buffer B was 80% acetonitrile, 0.1% formic acid. Peptides were separated on a segmented gradient from 2% to 5% buffer B for 10 min, from 5% to 20% buffer B for 100 min, from 20% to 25% buffer B for 10 min, and from 25% to 45% buffer B for 10 min at 200 nl / min (EASY-nLC 1000). Using the EASY-nLC 1200 system, peptides were separated on a segmented gradient from 3% to 6% buffer B for 10 min, from 6% to 25% buffer B for 100 min, from 25% to 31% buffer B for 10 min, and from 31% to 60% buffer B for 10 min, at 200 nl / min. Eluting peptides were analyzed on a QExactive HF mass spectrometer (Thermo Fisher Scientific). Peptide precursor mass to charge ratio (m/z) measurements (MS1) were carried out at 60000 resolution in the 300 to 1800 m/z range. The top ten most intense precursors with charge state from 2 to 7 only were selected for HCD fragmentation using 25% collision energy. The m/z of the peptide fragments (MS2) were measured at 30000 resolution using an AGC target of 2e5 and 80 ms maximum injection time. Upon fragmentation precursors were put on an exclusion list for 45 s. Peptides from the five knockout strains were analyzed in two runs and *Polrmt* knockout was used to control for the variability within runs. The time course analysis on *Lrpprc* knockout was performed in a single run.

## LC-MS/MS data analysis

The raw data were analyzed with MaxQuant version 1.4.1.2 (RRID:SCR_014485; *Cox and Mann, 2008*) using the integrated Andromeda search engine (*Cox et al., 2011*). Peptide fragmentation spectra were searched against the canonical and isoform sequences of the mouse reference proteome (proteome ID UP000000589, August 2015 from UniProt). The database was complemented with 245 sequences of contaminating proteins by MaxQuant. For the analysis methionine oxidation and protein N-terminal acetylation were set as variable modifications. The digestion parameters were set to 'specific' and 'Trypsin/P,' allowing for cleavage after lysine and arginine also when followed by proline. The minimum number of peptides and razor peptides for protein identification was 1; the minimum number of unique peptides was 0. Protein identification was performed at a peptide spectrum matches and protein false discovery rate (FDR) of 0.01. The 'second peptide' option was on to identify co-fragmented peptides. Successful identifications were transferred between the different raw files using the 'Match between runs' option, using a match time window of 0.7 min. Label-free quantification (LFQ) (*Cox et al., 2014*) was performed using an LFQ minimum ratio count of 2.

## Protein quantification analysis

Analysis of the label-free quantification (LFQ) results was done using the Perseus computation platform (*Tyanova et al., 2016*), version 1.5.0.0 and R, version 3.3.0 (*R Core Team, 2010*). For the analysis, LFQ intensity values were loaded in Perseus and all identified proteins marked as 'Reverse', 'Only identified by site', and 'Potential contaminant' were removed. The corresponding L/L and the L/L, cre genotypes were loaded separately in Perseus, the LFQ intensity values were log2 transformed and all proteins that contained less than two to five missing values in one of the groups (L/L or L/L, cre) were removed (*Supplementary file 1*). Missing values in the resulting subset of proteins were imputed with a width of 0.3 and down shift of 1.8. Imputed LFQ intensities were loaded into R where a two-sided moderated t-test analysis was performed using limma, version 3.30.13 (*Ritchie et al., 2015*). Proteins with an adjusted p value ('BH' correction) of less than 0.05 were

designated as differentially expressed. Our list of differentially expressed proteins was combined with annotations from MitoCarta2.0 (*Calvo et al., 2016*). The first of the semicolon separated entries in the 'Gene names' column was used to merge annotations through the 'Symbol' or 'Synonyms' columns of the Mouse.MitoCarta2.0.txt file. For protein entries that lacked a 'Gene names' entry, gene name information was retrieved using UniProt.ws, version 2.14 (*Carlson, 2017*). Using the principles of the 'Total Protein approach' (*Wiśniewski et al., 2014*), the LFQ intensity values were used to calculate the contribution of all mitochondrial proteins, according to MitoCarta2.0, to the total protein mass. Log2 transformed LFQ intensity and fold changes were used to generate density ad MA-plots plots. The mean LFQ value of controls was used to generate MA-plots.

Mouse MitoCarta2 protein sequences (Mouse.MitoCarta2.0.fasta) were downloaded from https://www.broadinstitute.org/scientific-community/science/programs/metabolic-disease-program/publications/mitocarta/mitocarta-in-0. The sequences were combined with the Mouse Mitocarta2 protein annotations using the RefSeq ID and the protein length. In cases of multiple matches with identical sequences, a single sequence was randomly selected. The extracted sequences were analyzed using TargetP 1.1 (*Emanuelsson et al., 2000*). The analysis was performed using 'Non-plant' and 'no cut-offs' parameters. The 'Perform cleavage site predictions' option (*Nielsen et al., 1997*) was enabled. Protein sequences predicted to localize to the Mitochondrion, indicated by 'M' in the Loc column, with reliability class (RC) of one, two, or three, were N-terminally trimmed by the value of predicted mitochondrial target peptide (mTP) presequence length (TPlen columns). The initial methionine was removed from the remaining of sequences. In addition, a second set of sequences was derived by removing the initial methionine only. The seqinr package (*Charif and Lobry, 2007*) was used to calculate the isoelectric points (pI) and the grand average of hydropathy (GRAVY) scores of the two sets of sequences. The GRAVY scores were computed by calculating the sum of the Hydropathy index (*Kyte and Doolittle, 1982*) of all amino acids in the protein sequence and dividing the value by the protein sequence length. The pI and GRAVY score values were combined with the average log2 RNA reads count per L/L, cre genotype and with the number of genotypes that had quantification values in the proteomic analysis (see *Supplementary file 7*). The data was divided into two disjoint sets, namely proteins quantified in zero genotypes and proteins quantified in at least one genotype. The empirical cumulative distributions of the pIs, GRAVY scores and the average log2 RNA counts of the two sets were visualized using the stat_ecdf function of the ggplot2 package.

Mitochondrial proteins were manually placed in functional categories based on literature and Uniprot (*The UniProt Consortium, 2017*). For the L/L comparison across different time points, MaxQuant's proteinGroups.txt file was combined with annotations from MitoCarta2.0 (*Calvo et al., 2016*) as described above. The MitoCarta annotated data were imported into Perseus and filtered for at least nine valid LFQ intensity values across all samples (26 in total). Only proteins annotated as mitochondrial, according to the 'MitoCarta2_List' column, were kept for analysis. Missing values were imputed as described above. Limma's F-statistic was used to detect proteins with significant differences in protein expression across the time points. Testing was performed after the data were grouped per week (nine week groups in total). The fold changes of the average LFQ intensity for all time points, relative to week three were calculated, scaled and z-score normalized in R. Hierarchical clustering of the normalized data was performed with pheatmap, version 1.0.8 (*Kolde, 2015*), using Euclidean distance as a metric. The rows (proteins) dendrogram was cut into to ten clusters and proteins, which partitioned into the four main clusters were used. Profiles plots of the normalized fold change values were color-coded using the distance of each profile from the cluster center (mean value).

For the *Lrpprc* knockout time course analysis data were imported into Perseus, filtered for at least 23 valid LFQ intensity values across all samples (46 in total), and missing values were imputed as described above. Limma's F-statistic was used to detect proteins with significant differences in protein expression across the time-points for either the L/L or the L/L, cre genotype. In addition, the F-statistic was used to identify differentially expressed proteins that have a significant difference in the profile across the time-points. To compare complete categories of proteins (*Figure 7E*), Wilcoxon signed-ranked test followed by FDR correction was performed. p values < 0.05 were considered significant. Data was plotted using R packages pheatmap version 1.0.8 and ggplot2 version 2.2.1.

## 2D annotation enrichment analysis

2D annotation enrichment analysis of the proteomic and transcriptomic data was performed with Perseus (*Cox and Mann, 2012*) using Benjamini-Hochberg FDR for truncation and a threshold value of 0.2. For the evaluation of the protein abundances during the post-natal development, identification of overrepresented functional categories in separate clusters was performed using Fisher exact test in Perseus. Benjamini-Hochberg FDR was for truncation with a threshold value of 1. The top three overrepresented categories in each cluster, based on adjusted p value, are shown.

## RNA isolation, qRT-PCR and RNA sequencing

RNA was isolated from cultured cells and frozen mouse heart using the miRNeasy Mini Kit (Qiagen, Germany) and the concentration, purity, and integrity were confirmed using a Bioanalyzer. Reverse transcription experiments were carried out after cDNA synthesis (High Capacity cDNA reverse transcription kit, Applied Biosystems, Germany). qRT-PCR was performed using the Taqman 2x Universal PCR mastermix, No Amperase UNG (Applied Biosystems) and commercially available Taqman assays for mouse *Adck3 (Coq8a)*, *Adck4 (Coq9b)*, *Atf4*, *Myc*, *Coq2*, *Coq4*, *Coq5*, *Coq7*, *Fdps*, *Gabpa*, *Hmgcs1*, *Hmgcr*, *Mterf4*, *Nrf1*, *Pdss1*, *Pdss2*, *Pgc1a*, *Polrmt*, *Tfam* and *Twnk* (Life technologies, Germany). The quantity of transcripts was normalized to *Beta-2-microglobulin* (*B2m*) as reference gene transcript. RNA sequencing was performed on total RNA on the Illumina HiSeq platform according to the Illumina Tru-Seq protocol. Random hexamer primers were used for cDNA library generation and carried out cytoplasmic rRNA depletion using the Ribo-Zero rRNA removal kit. The alignment to the *Mus musculus* reference genome (GRCm38) was performed using HISAT2 version 2.0.4 (hisat2 -p 8 –dta; *Kim et al., 2015*). Alignment files were sorted and indexed with SAMtools version 1.3.1 (samtools sort -@ 8; samtools index -b; *Li et al., 2009*). The transcript abundances were estimated with StringTie 1.2.4 (stringtie -p 18 -e –B –G) using the Ensembl 81 annotation (*Pertea et al., 2016*). The raw reads count matrix at gene-level was extracted with prepDE.py script provided at http://ccb.jhu.edu/software/stringtie/dl/prepDE.py on Python version 2.7.6. Differential expression analyses were performed with DESeq2 package R version 3.3.2 according to the standard DESeq2 tutorial that is part of the package (*Love et al., 2014*). Genes with an adjusted $p<0.05$ were considered statistically significant. Genes were placed manually in functional categories as described above and heatmaps were generated using pheatmap R package version 1.0.8. Mitochondrial-encoded RNAs were analyzed according to the same protocol except the NUMT regions were masked, filtered to reads in proper pairs and analyzed by DESeq2 as described above.

## RNA-Seq data analysis

RNA-Seq data were analyzed through the use of Ingenuity Pathways Analysis (Ingenuity Systems, www.ingenuity.com, RRID: SCR_008653). Log2 transformed expression values and adjusted p values were uploaded to the program. Genes from the data set that met the adjusted p value cutoff of 0.05 in all the knockout strains of this study and were associated with a canonical pathway in Ingenuity's Knowledge Base were considered for the analysis. The significance of the association between the data set and the canonical pathway was measured in two ways: (1) A ratio of the number of molecules from the data set that map to the pathway divided by the total number of molecules that map to the canonical pathway is displayed, (2) Fisher's exact test was used to calculate a p value determining the probability that the association between the genes in the dataset and the canonical pathway is explained by chance alone. Genes from the RNA-Seq data sets that had a p value < 0.05 for all the knockout strains were loaded into Cytoscape (version 3.5.0; *Shannon et al., 2003*) and used as queries to the iRegulon plug-in (*Janky et al., 2014*). The putative regulatory region was selected to be 20 kb centered on the transcriptional start site (TSS), the normalized enrichment score (NES) threshold was set at 3.0, and the maximum false discovery rate (FDR) on motif similarity was set at 0.001. Transcription factor enrichment analyses were performed based on motif or Chip-seq peaks enrichment using the 10K (9713 PWMs) motif collection and the 1120 Chip-seq track collection, respectively. Target genes for MYC and ATF4 were obtained from published Chip-seq experiments (*Han et al., 2013*; *Seitz et al., 2011*).

## Determination of coenzyme Q and amino acids content

Quinones were extracted from 20 mg grinded frozen mouse hearts as previously described for MEFs and quinone quantifications were performed as reported previously with the following changes: standard curve range was 4–3500 ng/mL for Q10 and 100–2800 ng/mL for Q9 (*Mourier et al., 2015*). Other metabolites analyzed in this study were extracted from 20 to 30 mg grinded frozen mouse hearts in 1 ml sample buffer (methanol:$H_2O$:chloroform in a 7:2:1 ratio). Samples were mixed, sonicated (10 cycles per 30 s) and centrifuged at 15000 g at 4°C for 5 min. Supernatants were centrifuged through 0.25 µm Centrifugal filters (VWR, Germany) and re-centrifuged twice at 15000 g for 5 min, placed at −80°C for 2 hr and re-centrifuged. Supernatants were dried in a SpeedVac apparatus (Eppendorf concentrator plus 5305) at 45°C for 1 hr. Next, samples were suspended in 100 µl and diluted 1/20 in 5 mM ammonium formate +0.15% formic acid (Sigma-Aldrich, Germany), mixed, sonicated for 2 min and filtrated through a 0.2 µm modified nylon centrifugal filter (VWR). For absolute quantification in positive and negative ESI MRM (multi reaction monitoring) mode an Acquitiy UPLC I-Class System (Waters) was connected to a XevoTM TQ-S (Waters). A Discovery HS F5-3 (Supelco) 3 µm, 2.1 × 100 mm column was used at 25°C. Solvent A was 5 mM ammonium formate + 0.15% formic acid and B was acetonitrile (VWR). A gradient from 100% A to 80% in 3 min, to 4 min isocrate, to 4.5 min 50%, to 7 min 0%, to 10 min isocrate at a flow rate of 0.35 ml/min was used. An equilibration step of 7 min was performed after each injection. The dilutions were injected twice of each injection volume, 0.1 µL and 8 µl. The sample manager was set to 8°C and the source temperature to 150°C, desolvation temperature was 500°C and desolvation gas was set to 800 L/h, cone gas to 150 L/h. The following MRM transitions were used as quantifier for sarcosine m/z 89.84 to 44.02 cone 14V collision 18V, glycine m/z 75.9 to 30.08 cone 28V collision 6V and in a negative mode serine m/z 105.9 to 42.09 cone 72V collision 10V. The MassLynx (Waters) software was used for data management and TargetLynx (Waters) was used for data evaluation and absolute quantification. All compounds were freshly prepared and dissolved in 5 mM ammonium formate + 0.15% formic acid. An external standard calibration curve was calculated using 11 concentrations from 100 to 20000 ng/ml (all of them were prepared from stock solutions 100 µg/ml). Correlation coefficient: r < 0.990, the peak integrations were corrected manually, if necessary. Quality control standards of each standard were used during sample analysis and showed between 0.5% and 40% deviation respectively. Blanks after the standards, quality control and sample batch proved to be sufficient. No carry over was detected.

## Mitochondrial enzyme activity measurements

Heart tissue homogenates were prepared from 20 to 30 mg grinded frozen mouse hearts using the CelLytic MT Cell Lysis Reagent (Sigma-Aldrich) supplemented with EDTA-free complete protease inhibitor cocktail (Roche). Total protein concentration was determined in triplicate using the Lowry method (DC Protein Assay, BioRAD, Germany) and the protein concentration of all samples was equalized. To follow citrate synthase activity, increase in absorbance at 412 nm was recorded after the addition of 0.3 mM acetyl–coenzyme A, 0.5 mM oxaloacetate, and 0.1 mM 5,5′-dithiobis-2-nitrobenzoic acid using the Citrate Synthase Assay Kit (Sigma-Aldrich) in a Tecan Infinite pro M200 microplate reader instrument.

## Total protein isolation, isolation of mitochondrial extracts, western blot and antisera

Total proteins were isolated from frozen mouse hearts as previously described (*Ekstrand et al., 2004*). Mitochondria were isolated from mouse heart using differential centrifugation as previously reported (*Milenkovic et al., 2013*). Proteins were separated by SDS-PAGE (using 4–12% or 10% precast gels from Invitrogen) and then transferred onto polyvinylidene difluoride membranes (GE Healthcare, Germany). Immunodetection was performed according to standard techniques using enhanced chemiluminescence (Immun-Star HRP Luminol/Enhancer from Bio Rad). The following antibodies were used: ALDHA8A1 (#PA5-19392, Thermofisher Scientific, RRID: AB_10985670), CLPP (#WH0008192M1, Sigma-Aldrich, RRID: AB_1840782), COX4 (#4850, Cell Signaling, Germany, RRID: AB_2085424), CS (#ab129095, Abcam, UK, RRID: AB_11143209), FDPS (#ab189874, Abcam, RRID: AB_2716301), GLS (#ab93434, Abcam, RRID: AB_10561964), HMGCS1 (#ab194971, Abcam, RRID: AB_2716299), HSPA9 (#ab82591, Abcam, RRID: AB_1860633), LONP1 (#ab103809, Abcam, RRID:

AB_10858161), LRPPRC (*Ruzzenente et al., 2012*); RRID: AB_2716302), MRPL37 (#HPA025826, Sigma-Aldrich, RRID: AB_1854106), MRPL44 (#16394–1-AP, Proteintech, Germany, RRID: AB_2146062), MRPS35 (#16457–1-AP, Proteintech, RRID: AB_2146521), MTHFD1 (#ab103698, Abcam, RRID: AB_10862775), MTHFD2 (#ab37840, Abcam, RRID: AB_776544), NDUFA9 (#ab14713, Abcam, RRID: AB_301431), POLRMT (*Kühl et al., 2014*); RRID: AB_2716297), PYCR1 (#13108–1-AP, Proteintech, RRID: AB_2174878), SDHA (#459200, Thermofisher Scientific, RRID: AB_2532231), SHMT2 (#HPA020543, Sigma-Aldrich, RRID: AB_1856833), TFAM (#ab131607, Abcam, RRID: AB_11154693), Total OXPHOS Rodent WB Antibody Cocktail (#ab110413, Abcam, RRID: AB_2629281), Tubulin (clone 11H10, #2125, Cell Signaling, RRID: AB_2619646), TWINKLE (*Milenkovic et al., 2013*); RRID: AB_2716298), UQCRFS1 (#ab131152, Abcam, RRID: AB_2716303), VDAC1 (clone N152B/23 #MABN504, Millipore, Germany, RRID: AB_2716304).

## DNA isolation and mtDNA quantification

Total DNA was isolated from mouse heart tissue using the Puregene Cell and Tissue Kit (Qiagen). mtDNA was measured by semiquantitative PCR carried out on 4 ng of total DNA in a 7900HT Real-Time PCR System (Applied Biosystems) using TaqMan probes specific for the mt-*Co1*, mt-*Cytb*, mt-*Nd1*, mt-*Nd6*, and 18S genes (Applied Biosystems).

## Experimental design and statistical analysis

Sample sizes were defined based on the previous molecular characterization of the mouse strains described in *Table 1*. Each mouse was considered an independent biological replicate (n) and repeated measurements of the same biological replicate were considered technical replicates. $\geq 3$ biological replicates of each knockout strain (*Twnk*, *Tfam*, *Polrmt*, *Lrpprc* and *Mterf4*) and their respective age-matched control mice were used for all the experiments presented. Age and genotype were the only inclusion criteria considered. Statistical analyses for RNA-Seq and MS protein quantification analyses were performed as described above. The exact n and p values for the transcriptomic and proteomic analyses can be found in *Supplementary file 1–4*. For qRT-PCR and metabolomics analyses variance was assessed using an F-test and statistical significance was assessed by a two-tailed unpaired Student's *t*-test in Excel. The exact n and p values for metabolomics, qRT-PCR analyses, and enzyme activity measurements can be found in *Supplementary file 5*. Two to three technical replicates were performed for targeted metabolomics measurements, three technical replicates were performed for qRT-PCR and semiquantitative PCR analyses, three to seven technical replicates were performed for citrate synthase activity measurements. Proteomic, transcriptomic, qRT-PCR experiments were performed once. Metabolomic and semiquantitative PCR mtDNA quantifications determinations were performed two times. Western blot analyses were repeated >3 times with the exception of the data presented in *Figure 5—figure supplement 2*. Sample processing for RNA-Seq, qRT-PCR, semiquantitative PCR, proteomics, enzyme activity measurements, and metabolomics analyses was performed blindly. Pearson correlation analysis of transcriptomics and proteomics across the different lines were performed and visualized using pheatmap R package version 1.0.8. The definition of center and precision measures, and p values are provided in the figure legends. $p < 0.05$ was considered significant.

## Data availability

Raw RNA-Seq data have been deposited in the Gene Expression Omnibus repository under accession number GSE96518. The proteomics datasets presented are available in *supplementary File 1–3*, *8* and *9*.

## Acknowledgements

We thank Dr. Xinping Li, Proteomics Core Facility, Max Planck Institute for Biology of Ageing, Nadine Hochhard and Lysann Schmitz for technical assistance. RNA library construction and sequencing were performed at the Cologne Center for Genomics.

## Additional information

### Funding

| Funder | Grant reference number | Author |
|---|---|---|
| Australian Research Council | DP170103000 | Aleksandra Filipovska |
| National Health and Medical Research Council | APP1067837 | Aleksandra Filipovska |
| National Health and Medical Research Council | APP1058442 | Aleksandra Filipovska |
| H2020 European Research Council | Advanced Investigator Grant (268897) | Nils-Göran Larsson |
| Max-Planck-Gesellschaft | | Nils-Göran Larsson |
| Vetenskapsrådet | 2015-00418 | Nils-Göran Larsson |
| Knut och Alice Wallenbergs Stiftelse | | Nils-Göran Larsson |

The funders had no role in study design, data collection and interpretation, or the decision to submit the work for publication.

### Author contributions

Inge Kühl, Conceptualization, Data curation, Formal analysis, Supervision, Validation, Investigation, Visualization, Methodology, Writing—original draft, Project administration, Writing—review and editing; Maria Miranda, Conceptualization, Data curation, Formal analysis, Validation, Investigation, Visualization, Methodology, Writing—original draft, Writing—review and editing; Ilian Atanassov, Data curation, Analysis of proteomic data, Integrated analysis of transcriptomic and proteomic data, Investigation, Validation, Visualization; Irina Kuznetsova, Analysis of transcriptomic data, Visualization; Yvonne Hinze, Investigation; Arnaud Mourier, Methodology; Aleksandra Filipovska, Formal analysis; Nils-Göran Larsson, Resources, Supervision, Funding acquisition, Writing—original draft, Project administration, Writing—review and editing

### Author ORCIDs

Inge Kühl ![iD] http://orcid.org/0000-0003-4797-0859
Maria Miranda ![iD] http://orcid.org/0000-0002-0817-553X
Ilian Atanassov ![iD] https://orcid.org/0000-0001-8259-2545
Aleksandra Filipovska ![iD] http://orcid.org/0000-0002-6998-8403
Nils-Göran Larsson ![iD] http://orcid.org/0000-0001-5100-996X

### Ethics

Animal experimentation: The health status of the animals is specific pathogen free according to the Federation of the European Laboratory Animal Science Association (FELASA) recommendations. All animal procedures were conducted in accordance with European, national and institutional guidelines and protocols (no.: AZ.: 84-02.05.50.15.004 and AZ.: 84-02.04.2015.A103) were approved by the Landesamt für Natur, Umwelt und Verbraucherschutz, Nordrhein-Westfalen, Germany.

### Decision letter and Author response

Decision letter https://doi.org/10.7554/eLife.30952.045
Author response https://doi.org/10.7554/eLife.30952.046

## Additional files

### Supplementary files

• Supplementary file 1. Comparative analysis of mitoproteomic data from heart of *Tfam, Twnk, Polrmt, Lrpprc* and *Mterf4* knockout mice and corresponding controls.

DOI: https://doi.org/10.7554/eLife.30952.033

• Supplementary file 2. Analysis of mitoproteomic data from heart at different ages of *Lrpprc* knock-out mice and controls.
DOI: https://doi.org/10.7554/eLife.30952.034

• Supplementary file 3. Analysis of mitoproteomic data from heart at different ages of control mouse strains.
DOI: https://doi.org/10.7554/eLife.30952.035

• Supplementary file 4. Analysis of total cellular transcriptome from heart of *Tfam, Twnk, Polrmt, Lrpprc* and *Mterf4* knockout and control mouse strains at different ages.
DOI: https://doi.org/10.7554/eLife.30952.036

• Supplementary file 5. Number of biological replicates and p values of qRT-PCR, metabolomic analyses and enzyme activity measurements.
DOI: https://doi.org/10.7554/eLife.30952.037

• Supplementary file 6. iRegulon analysis of RNA-Seq data of total RNA from hearts of end-stage conditional knockout mice.
DOI: https://doi.org/10.7554/eLife.30952.038

• Supplementary file 7. Analysis of proteomic bias in mitoproteomics data from heart of *Tfam, Twnk, Polrmt, Lrpprc* and *Mterf4* knockout mice and corresponding controls.
DOI: https://doi.org/10.7554/eLife.30952.039

• Supplementary file 8. Complete set of differential expression proteomic analysis in heart of the five knockout mouse strains and according controls; boxplots of the intensity detected by mass spectrometry per protein.
DOI: https://doi.org/10.7554/eLife.30952.040

• Supplementary file 9. Complete set of sequential mitoproteomic changes at different time points of progressive mitocondrial dysfunction in heart of one knockout mouse strain. Time curves of differential expression analysis of each protein on the *Lrpprc* knockout analysis at different ages.
DOI: https://doi.org/10.7554/eLife.30952.041

• Transparent reporting form
DOI: https://doi.org/10.7554/eLife.30952.042

## Major datasets

The following dataset was generated:

| Author(s) | Year | Dataset title | Dataset URL | Database, license, and accessibility information |
|---|---|---|---|---|
| Kühl I, Miranda M, Atanassov I, Kuznetsova I, Hinze Y, Mourier A, Filipovska A, Larsson NG | 2017 | Transcriptomic and proteomic landscape of mitochondrial dysfunction reveals secondary coenzyme Q deficiency in mammals | https://www.ncbi.nlm.nih.gov/geo/query/acc.cgi?acc=GSE96518 | Publicly available at the NCBI Gene Expression Omnibus (accession no: GSE96518) |

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
