## [Decision Letter]

Thank you for submitting your article "Transcriptomic and proteomic landscape of mitochondrial dysfunction reveals secondary coenzyme Q deficiency in mammals" for consideration by *eLife*. Your article has been favorably evaluated by a Senior Editor and three reviewers, one of whom, Agnieszka Chacinska (Reviewer #1), is a member of our Board of Reviewing Editors. The following individual involved in review of your submission has agreed to reveal their identity: Antoni Barrientos (Reviewer #3).

The reviewers have discussed the reviews with one another and the Reviewing Editor has drafted this decision to help you prepare a revised submission.

In this manuscript by the Larsson's group, the authors report on global consequences of mtDNA expression dysfunction in the mouse heart. The comprehensive analysis, including transcriptomics, proteomics and metabolomics, of five mitochondrial pathology models with conditional heart-specific mutations in imitochondrial proteins is presented. Their main novel finding is related to the global down-regulation of CoQ biosynthetic enzymes, which appears to occur on a post-transcriptional level, as well as reduced tissue CoQ levels.

The reviewers found the study impressive and of high quality, ideally suited as a resource paper and of immense value for researchers and clinicians. Below are some constructive critiques on the presentation and interpretation of the data.

Essential revisions:

In terms of experiments, the reviewers agreed that protein abundance, inferred from MS, should be verified using ELISA or Western blot on a small handful of proteins, including the representatives of those proteins that change and do not change. Furthermore, citrate synthase activity (a parameter used in clinics) as a measure of mitochondrial abundance should be tested.

Several additional computational analyses need to be performed.

- Accuracy of protein fold-change inference.

Differential expression analyses typically identify a relatively small set of changing transcripts/proteins against the backdrop of a much larger set of unperturbed ones. In this setting, a change in the relative fraction of reads/spectra mapping to a transcript/protein can be assumed to derive primarily from a change in its own abundance rather than from the cumulative effects of changes in the abundance of many other transcripts/proteins.

However, the mito-proteome data presented here shows more than half of mito-localized proteins in almost every KO model are differentially expressed. Wholesale remodeling of the mitochondrial proteome is certainly biologically plausible in these KO models, but it raises some concerns about the accuracy of inferred fold-changes for individual proteins. In particular, the highly abundant OXPHOS proteins are strongly depleted in all KO models – drastically changing the composition of the mito-proteome and potentially elevating the fractional representation of remaining proteins with no underlying change in their abundance. The fact that authors identify sets of proteins that change coordinately in a consistent direction across KO models is reassuring, and suggests the main thrust of the signal they report is unlikely to be confounded. To increase readers' confidence that individual protein changes are accurate and comparable the authors should also show i) the distribution of protein abundances and the distribution of control/KO fold-changes for all proteins in each mouse model, ii) show plots relating the abundance of proteins and their fold-changes, to rule out that originally low-abundance proteins are artifactually inflated in KOs. An MA-plot could achieve this goal, but perhaps the x-axis should be the mean abundance in the control only rather than mean abundance across all samples.

- Systematic bias in protein sampling.

Authors should perform some analyses to check if their proteomics pipeline systematically misses or under-represents certain classes of proteins, such as low-abundance proteins, hydrophobic vs. charged, membrane-bound vs. free. The selection of column, the MS data capture sequence, etc. can influence this sampling. This can be easily done by taking the entire MitoCarta2.0 collection, and breaking it up into two disjoint sets, those MitoCarta2.0 proteins detected (set 1), and those MitoCarta2.0 proteins that are not detected (set 2). Then the cumulative distribution function (cdf) of pI, hydrophobicity, RNA-seq abundance, can be plotted for set 1 versus set 2. Such a plot will reveal any systematic biases in their proteomics datasets, again helping to showcase its strengths and weaknesses.

- Visualization of transcription factor targets.

The paper puts a prominent focus on the Atf4 and c-Myc transcription factors as orchestrators of the response to mitochondrial dysfunction but none of the figures breaking up genes into categories includes a category such as "Atf4 target" or "c-Myc target". Target annotations can be obtained from the literature (e.g., Han et al. 2013, for Atf4).

It seems likely that categories such as "apoptosis", "degradation and stress response", "tRNA charging", and "mitochondrial 1C pathway" – which happen to show the strongest transcript/protein correlation – all include numerous targets of these transcription factors. Thus, the key trend is driven by their joint transcriptional regulation rather than the disparate functional categories. Authors are encouraged to find a way to incorporate this into at least one of their "gene category" figures, and perhaps into the volcano plots as well.

Along this line of analysis, it would be interesting to discuss the consequences for the cell if these transcriptional responses would be abolished? Would the respiratory dysfunction be rather accelerated (because less of oxphos protein synthesis) or less severe, less oxphos protein synthesis or less "secondary responses", related to stress, such as the increase in mitochondrial proteases?

---

## [Author Response]

Essential revisions:In terms of experiments, the reviewers agreed that protein abundance, inferred from MS, should be verified using ELISA or Western blot on a small handful of proteins, including the representatives of those proteins that change and do not change.

We verified the steady-state levels of the following mitochondrial proteins by western blotting on mitochondrial extracts: ALDHA8A1, ATP5A, CLPP, COX4, CS, HSPA9, GLS, LRPPRC, LONP, MRPL37, MRPL44, MRPS35, MT-CO1, NDUFA9, NDUFB6, POLRMT, PYCR1, UQCRFS1, UQCRC2, TFAM and TWINKLE. All of the proteins tested show the same trend as found by our proteomic approach. Please note that the proteins that do not change are not listed in the heat maps of the main figures, but their expression level can be assessed in the box plots in the Supplementary file 8. We added the western blots to the new Figure 5—figure supplement 2 including a new figure legend. Information on the preparation of mitochondrial extracts, western blotting and the antibodies was added to the Materials and methods subsection “Total protein isolation, isolation of mitochondrial extracts, western blot and antisera” and to the Key Resources Table. We also updated the “Experimental design and statistical analysis” subsection in the Materials and methods.

Furthermore, citrate synthase activity (a parameter used in clinics) as a measure of mitochondrial abundance should be tested.

We assessed citrate synthase activity on heart tissue of the five knockout mouse models and according controls, and added this data to Figure 1—figure supplement 1 and figure legend. A significant increase of citrate synthase activity was obtained in *Tfam* and *Mterf4* knockout hearts consistent with previous publications (Hansson et al., 2004, Camara et al., 2011), whereas the other knockouts had normal citrate synthase activity. It is important to recognize that our study is about responses induced by progressive respiratory chain dysfunction caused by different knockouts of genes critical for mtDNA gene expression. It is interesting in this respect that the induction of citrate synthase activity is more pronounced in some knockouts than in other. Citrate synthase activity is typically used as a measure for mitochondrial abundance, but it is only one of several markers. Unfortunately, the different maximal life span of the knockouts makes it difficult to use citrate synthase for comparison between the strain, e.g. the *Mterf4* knockout has the longest life span and the highest levels of citrate synthase activity at the end stage. The available time for induction of a robust mitochondrial biogenesis response is thus likely an important factor. To avoid confusing the readers, we have decided to down-tune the statements on mitochondrial biogenesis and to add additional information, by:

Removing the row “biogenesis” from Table 1;

Adding a sentence stating: “Two of the mouse models showed an increase in citrate synthase activity which is a commonly used marker of mitochondrial biogenesis (Figure 1—figure supplement 1).”;

Modifying the sentence so that it now states: “Therefore, it is possible that MYC contributes to the metabolic rewiring that is caused by mitochondrial dysfunction”;

Adding the description on the experimental measurement of citrate synthase activity to the Materials and methods subsection “Mitochondrial enzyme activity measurements”;

Updating the “Experimental design and statistical analysis” subsection in the Materials and methods, including the addition of statistical analysis information for the citrate synthase measurements to Supplementary file 5.

Several additional computational analyses need to be performed.- Accuracy of protein fold-change inference.Differential expression analyses typically identify a relatively small set of changing transcripts/proteins against the backdrop of a much larger set of unperturbed ones. In this setting, a change in the relative fraction of reads/spectra mapping to a transcript/protein can be assumed to derive primarily from a change in its own abundance rather than from the cumulative effects of changes in the abundance of many other transcripts/proteins.However, the mito-proteome data presented here shows more than half of mito-localized proteins in almost every KO model are differentially expressed. Wholesale remodeling of the mitochondrial proteome is certainly biologically plausible in these KO models, but it raises some concerns about the accuracy of inferred fold-changes for individual proteins. In particular, the highly abundant OXPHOS proteins are strongly depleted in all KO models – drastically changing the composition of the mito-proteome and potentially elevating the fractional representation of remaining proteins with no underlying change in their abundance. The fact that authors identify sets of proteins that change coordinately in a consistent direction across KO models is reassuring, and suggests the main thrust of the signal they report is unlikely to be confounded. To increase readers' confidence that individual protein changes are accurate and comparable the authors should also show i) the distribution of protein abundances and the distribution of control/KO fold-changes for all proteins in each mouse model, ii) show plots relating the abundance of proteins and their fold-changes, to rule out that originally low-abundance proteins are artifactually inflated in KOs. An MA-plot could achieve this goal, but perhaps the x-axis should be the mean abundance in the control only rather than mean abundance across all samples.

We have generated the distribution plots for the Log2 label-free quantification (LFQ) intensity determined by mass spectrometry for all quantified proteins, the distribution plots for the fold-changes of knockout (L/L, cre) vs. controls (L/L), and the MA-plots showing the relationship between log2 fold-changes vs. average LFQ intensity of controls (L/L), knockouts (L/L, cre), and both per knockout mouse strain. The figures are presented in the Figure 1—figure supplement 4, and a figure legend explaining these new panels has been added.

Overall, we do not expect a big inflation of ratios since there are both, up and down regulated proteins with similar magnitude in all of the knockout mouse models and the perturbation of the mitoproteomes are significantly different for *Twinkle, Tfam* and *Polrmt* vs. *Lrpprc* and *Mterf4* knockouts due to the abundance of mitoribosomal proteins. However, the MA-plots revealed indeed that some of the proteins with higher log2 fold-changes in all the mouse strains correspond to proteins with lower LFQ intensity and their fold changes might thus be inflated. Among these proteins are ALDH1L2, ALDH18A1, MTHFD2, MAOA and PYCR1. Consistent with the low LFQ intensities and marked upregulation, these proteins were detected in all the knockout samples but the controls have many imputed values (see Supplementary file 8). We verified the protein levels of ALDH18A1, MTHFD2 and PYCR1 by immunoblots on mitochondrial extracts (see our response to point 1 and Figure 5—figure supplement 2, Figure 6) and found a consistent increase in the protein levels giving confidence on the filtering criteria that were used to analyse the mitoproteomic data. We thank the reviewers for the suggestion to perform these analyses and feel that the added Figure 1—figure supplement 4 will aid the readers in the interpretation of the data presented in our paper.

- Systematic bias in protein sampling.Authors should perform some analyses to check if their proteomics pipeline systematically misses or under-represents certain classes of proteins, such as low-abundance proteins, hydrophobic vs. charged, membrane-bound vs. free. The selection of column, the MS data capture sequence, etc. can influence this sampling. This can be easily done by taking the entire MitoCarta2.0 collection, and breaking it up into two disjoint sets, those MitoCarta2.0 proteins detected (set 1), and those MitoCarta2.0 proteins that are not detected (set 2). Then the cumulative distribution function (cdf) of pI, hydrophobicity, RNA-seq abundance, can be plotted for set 1 versus set 2. Such a plot will reveal any systematic biases in their proteomics datasets, again helping to showcase its strengths and weaknesses.

We have generated the cumulative distribution function plots including all the MitoCarta2 proteins that were detected versus not-detected and analyzed against potential bias for hydrophobicity, charge and abundance. We defined detected proteins as quantified MitoCarta2 proteins detected in at least one knockout mouse strain, in other words, proteins that fulfilled the filtering criteria as described in the Materials and methods subsection “Protein quantification analysis” and the Supplementary file 1. These analyses revealed a bias towards abundant proteins, which is a common limitation for shot-gun mass spectrometry approaches. However, there were not big differences with regard to hydrophobicity and charge, in particular when the sequence of the mitochondrial targeting peptide is removed. We have added a description of these analyses to the aforementioned Materials and methods subsection. The results have been included in the subsection “Integrated RNA-sequencing and label-free mass spectrometry approach identify cellular and mitochondrial responses to OXPHOS dysfunction in mouse”, second paragraph and Figure 1—figure supplement 3. The data used to estimate protein bias are included in the new Supplementary file 7.

- Visualization of transcription factor targets.The paper puts a prominent focus on the Atf4 and c-Myc transcription factors as orchestrators of the response to mitochondrial dysfunction but none of the figures breaking up genes into categories includes a category such as "Atf4 target" or "c-Myc target". Target annotations can be obtained from the literature (e.g., Han et al. 2013, for Atf4).

We did transcription factor enrichment analysis on the common significantly regulated genes (adjusted p <0.05) across the five knockout mouse strains using the Cytoscape plugin iRegulon. Both, *Atf4* and *Myc*, are amongst the predicted enriched transcription factors based on ChIP-seq tracks or Motif collections. We generated heatmaps representing the fold changes of the expression of the ATF4 and MYCtarget genes (obtained from Han et al., 2013 and Seitz et al., 2011, respectively) in the 5 knockout mouse hearts compared to controls. The mitochondrial target genes are illustrated in the new Figure 3 and the total number of target genes is shown in the new Figure 3—figure supplement 1. According new figure legends were added. The data supporting the iRegulon analysis were added to a new Supplementary file 6 and the results of the analyses were added to the Results subsection “Atf4 and Myc transcription factors are increased in mouse hearts with severe mitochondrial dysfunction”.

The analysis method is included in the Materials and methods section. The subsection title “Canonical pathway analysis of RNA-Seq data” was changed to “RNA-Seq data analysis”. Text explaining the new bioinformatic analyses was added to this section. Furthermore, we changed the wording “and significant changes in several MYC target genes” to “Remarkably, all knockouts in our study show a drastic increase of *Myc* transcripts and significant changes in several MYC target genes which supports the idea that MYC is involved in remodeling mitochondrial metabolism under severe mitochondrial dysfunction.”

It seems likely that categories such as "apoptosis", "degradation and stress response", "tRNA charging", and "mitochondrial 1C pathway" – which happen to show the strongest transcript/protein correlation – all include numerous targets of these transcription factors. Thus, the key trend is driven by their joint transcriptional regulation rather than the disparate functional categories. Authors are encouraged to find a way to incorporate this into at least one of their "gene category" figures, and perhaps into the volcano plots as well.

We agree that the genes that show the strongest transcript/protein correlation include some of the targets of ATF4/MYC. Thus, we incorporated the analysis of transcription regulation by the two transcription factors. We coloured the mitochondrial target genes of eitherATF4 or MYC in the scatterplots of the RNA-Seq versus the proteomics data of all the knockouts of the different categories. We show this data for the categories that have clear changes at the transcript level such as apoptosis, degradation and stress response, mitochondrial 1 C pathway, and pyruvate and amino acid metabolism in the new Figure 4—figure supplement 3. Although our data shows a clear correlation between their target genes and some of our categories, we do not detect the same transcriptional response in all of the target genes. Since the transcriptional regulation via ATF4 and MYC is very complex we do not feel confident to draw any conclusions on a causal response based on the experiments we have performed. We have explained these issues by adding text to the subsection “Atf4 and Myc transcription factors are increased in mouse hearts with severe mitochondrial dysfunction” and subsection “Atf4 and Myc transcription factors are increased in mouse hearts with severe mitochondrial dysfunction”, sixth paragraph, as well as by adding panel C and D to Figure 3 and by creating the new Figure 3—figure supplement 1.

Along this line of analysis, it would be interesting to discuss the consequences for the cell if these transcriptional responses would be abolished? Would the respiratory dysfunction be rather accelerated (because less of oxphos protein synthesis) or less severe, less oxphos protein synthesis or less "secondary responses", related to stress, such as the increase in mitochondrial proteases?

Both, ATF4 and MYC, regulate several essential cellular processes during cell cycle and response to stress. Thus, abolishing the complete transcriptional responses of these transcription factors would have severe side effects for cell survival. However, the strong dysregulation of the mitochondrial proteome we detect in the different knockout mouse models with a primary defect in mtDNA gene expression and lack of mtDNA-encoded OXPHOS proteins, clearly shows that the secondary responses to stress contribute significantly to the altered mitochondrial proteome landscape upon mitochondrial dysfunction. Recently, Kahn and collaborators showed that in skeletal muscle mTORC1 acts upstream of ATF4 and regulates some of the stress responses upon mitochondrial dysfunction. Treatment with rapamycin, abolished the activation of these stress responses and improved the clinical phenotype of these mouse models indicating that controlling the secondary responses to stress would reduce the speed of the progression of respiratory dysfunction. Since we feel it is beyond the scope of our manuscript to perform experiments to address this question, we have instead included a discussion based on the available literature (Discussion, fourth paragraph).